# Concentrations and source regions of light absorbing particles in snow/ice in northern Pakistan and their impact on snow albedo

Chaman Gul[1,2,5], Siva Praveen Puppala[2], Shichang Kang[1,3,5], Bhupesh Adhikary[2], Yulan Zhang[1], Shaukat Ali[4], Yang Li[3], Xiaofei Li[1]

[1]State Key Laboratory of Cryosphere Science, Northwest Institute of Eco-Environment and Resources, Chinese Academy of Sciences, Lanzhou 73000, China
[2]International Centre for Integrated Mountain Development (ICIMOD), G.P.O. Box 3226, Kathmandu, Nepal
[3]CAS Center for Excellence in Tibetan Plateau Earth Sciences, Beijing, 100101, China
[4]Global Change Impact Studies Centre (GCISC), Ministry of Climate Change, Islamabad, Pakistan
[5]University of Chinese Academy of Sciences, Beijing, China

*Correspondence to*: Chaman Gul (chaman.gul@icimod.org; chaman@lzb.ac.cn), Siva Praveen Puppala (SivaPraveen.Puppala@icimod.org).

**Abstract.** Black carbon (BC), water-insoluble organic carbon (OC), and mineral dust are important particles in snow and ice, which significantly reduce albedo and accelerate melting. Surface snow and ice samples were collected from the Karakoram-Himalayan region of northern Pakistan during 2015 and 2016 in summer (six glaciers), autumn (two glaciers), and winter (six mountain valleys). The average BC concentration overall was $2130 \pm 1560$ ng g$^{-1}$ in summer samples, $2883 \pm 3439$ ng g$^{-1}$ in autumn samples, and $992 \pm 883$ ng g$^{-1}$ in winter samples. The average water insoluble OC concentration overall was $1839 \pm 1108$ ng g$^{-1}$ in summer samples, $1423 \pm 208$ ng g$^{-1}$ in autumn samples, and $1342 \pm 672$ ng g$^{-1}$ in winter samples. The overall concentration of BC, OC, and dust in aged snow samples collected during the summer campaign was higher than the concentration in ice samples. The values are relatively high compared to reports by others for the Himalayas and Tibetan Plateau. This is probably the result of taking more representative samples at lower elevation where deposition is higher and the effects of ageing and enrichment more marked. A reduction in snow albedo of 0.1–8.3% for fresh snow and 0.9–32.5% for aged snow was calculated for selected solar zenith angles during daytime using the Snow, Ice, and Aerosol Radiation (SNICAR) model. Daily mean albedo was reduced by 0.07–12.0%. The calculated radiative forcing ranged from 0.16 to 43.45 Wm$^{-2}$ depending on snow type, solar zenith angle, and location. The potential source regions of the deposited pollutants were identified using spatial variance in wind vector maps, emission inventories coupled with backward air trajectories, and simple region tagged chemical transport modelling. Central, South, and West Asia were the major sources of pollutants during the sampling months, with only a small contribution from East Asia. Analysis based on the Weather Research and Forecasting (WRF-STEM) chemical transport model identified a significant contribution (more than 70%) from South Asia at selected sites. Research into the presence and effect of pollutants in the glaciated areas of Pakistan is economically significant because the surface water resources in the country mainly depend on the rivers (the Indus and its tributaries) that flow from this glaciated area.

## 1 Introduction

Carbon is an essential component of atmospheric aerosols, where it appears in the form of black carbon (BC, also known as elemental carbon, EC), and organic carbon (OC). BC is emitted into the atmosphere from incomplete combustion of carbon-based fuels (mainly fossil fuels and biomass) (Jacobson, 2004), while OC can be directly emitted into or formed in the atmosphere. After deposition on snow and ice surfaces, BC particles significantly reduce the snow albedo (hemispheric reflectance) in the visible part of the electromagnetic spectrum, cause snow albedo feedback (Doherty et al., 2013), enhance solar radiation absorption (Warren and Wiscombe, 1980), and accelerate snow melting (Hansen and Nazarenko, 2004). BC, both in air and deposited on snow, is important in net positive forcing of the climate. Clean snow is one of the most reflective natural surfaces on Earth at the ultraviolet and visible wavelengths, while BC is the most efficient light-absorbing species in the visible spectral range (Horvarth, 1993). One ng $g^{-1}$ of BC has almost the same effect on albedo reduction as 100 ng $g^{-1}$ mineral dust at 500 nm wavelength (Warren et al., 1982). However, the exact amount of albedo reduction also depends on the refractive index, grain size, solar zenith angle (SZA), snow density, dust particle size and concentration, particle morphology, surface roughness, snow depth, liquid water content, snow shape, and topography (Wiscombe and Warren 1985). Albedo reduction usually results in amplification of the energy absorbed by dirty snow (Painter et al., 2010). An albedo feedback is triggered and amplified by deposition of impurities on the snow surface which reduces snow albedo thus accelerating melting and further reducing albedo (Doherty et al., 2013; Flanner et al., 2009). Albedo feedback is amplified by the presence of light-absorbing particles (Doherty et al., 2013). Studies conducted in Greenland showed that at visible wavelengths 10 ng $g^{-1}$ coarse-grained BC particles in aged snow and 40 ng $g^{-1}$ BC particles in new snow can reduce snow albedo by around 1 to 3% (Warren and Wiscombe, 1985).

Increased BC mass concentration and deposition on the Tibetan glaciers over the last 20 years (Xu et al., 2009a) have played a significant role in rapid glacier melting in the region (Xu et al., 2012; Yao et al., 2012). A high concentration of aerosol has deposited on the snow surface and increased the BC content in snow over the southern edge of the Tibetan Plateau to the north of the Himalayas (Gertler et al., 2016). The southern slope of the Himalayas is relatively even more exposed to BC due to emissions from India and transport through southwesterly and westerly winds (Xu et al., 2009; Yasunari et al., 2010). BC deposited on snow in the Himalayan region induces an increase in net shortwave radiation at the snow surface with an annual mean of about 1 to 3 $Wm^{-2}$, producing an estimated 0.05–0.3°C warming (Ménégoz et al., 2014). Deposition of anthropogenic BC has been observed to contribute significantly to the decrease in snow cover extent over recent decades (Dery et al., 2007), and shortening of the duration of the snow cover season by several days (Ménégoz et al., 2013a). The climate warming efficiency of BC in snow is greater than the warming efficiency of other anthropogenic pollutants, including carbon dioxide (Hansen et al., 2005). Another important characteristic of BC is its higher snowmelt efficiency. The snowmelt efficiency of BC in terms of snow cover fraction and snow water equivalent is larger than that induced by increase in carbon dioxide (Qian et al., 2011). The annual snow albedo reduction effect due to BC outweighs the aerosol dimming effect (reduction in solar radiation reaching the surface) by a factor of about six over the global snow cover (Flanner et al.,

2009).

At present, South and East Asia are considered to be the two largest BC emission regions in the world and likely

to remain so (Menon et al., 2010). BC transported from East Asia can be lifted high and moved towards the
northeast during the summer monsoon season (Zhang et al., 2015; Cong et al., 2015; Lüthi et al., 2015), affecting the
life of glaciers and snow-covered areas.

Research into the glaciers of the extended Himalayan region and Tibetan Plateau has prime importance because

these glaciers act as a water storage tower for South and East Asia, and shrinking could affect water resources for up
to a billion people (Immerzeel et al., 2010). The glaciated area in northern Pakistan may be more exposed to BC
effects than that in other regions because potentially it can receive emissions generated from both South and Central
Asia as well as from the Middle East. Meltwater coming from these glaciers flows into the river Indus, which has
major economic importance for the people of Pakistan.

A number of authors have described the concentration and impacts of light absorbing particles in the Tibetan

glaciers (for example Qian et al., 2015; Wang et al., 2015; Zhang et al., 2017; Li et al., 2017; Niu et al., 2017).
However, until now, no studies have been published related to the concentration of light absorbing aerosols in the
surface snow and ice of northern Pakistan, and although several authors have investigated transport pathways over
the Himalayan region (e.g. Babu et al., 2011 for the western trans-Himalayas; Lu et al., 2012 for the Tibetan Plateau
and Himalayas) little is known about the potential sources and transport pathways of pollutants affecting the
Pakistan area.

In this study, we looked at the concentration of light absorbing particles (BC, OC, dust) in snow and ice in

northern Pakistan, their impact on snow albedo and radiative forcing, and the likely source regions. Albedo was
estimated from the BC and dust concentrations identified in collected samples of snow and ice using the online snow
albedo simulation SNICAR model (Flanner et al., 2009). Radiative forcing was calculated from the albedo reduction
obtained from the SNICAR model together with the incident short-wave solar radiation obtained from the SBDART
(Santa Barbara DISORT Atmospheric Radiative Transfer) model. The frequency distribution of aerosol subtypes
(smoke, polluted continental, dust, and others) in the atmosphere over the study area was calculated for the snow and
ice sampling periods using CALIPSO (Cloud-Aerosol Lidar and Infrared Pathfinder Satellite Observations) satellite
data from 2006 to 2014 as a further indication of the types of aerosol contributing to the observed deposition. The
potential source regions of pollutants were identified using spatial variance in wind vector maps prepared using
MERRA-2 reanalysis data, calculation of back air trajectories using the HYSPLIT-4 (Hybrid Single Particle
Lagrangian Integrated Trajectory) model, and a simple region tagged chemical transport model (WRF-STEM). The
back air trajectories approach has been used in many studies to identify possible source regions for atmospheric and
deposited BC (Zhang et al., 2013). Pollutant source regions identified using the different approaches were compared
and the most likely source regions of the pollutants identified.

## 2 Methodology

### 2.1 Study area

The study area was located around 35.40°N 74.38°E in the mountains and adjacent mountain valleys of the Karakoram and Himalayan region in northern Pakistan (Figure 1). Snow and ice samples were collected in summer from six glaciers – Passu, Gulkin, Barpu, Mear, Sachin, and Henarche – and in autumn from Gulkin and Sachin (Figure 1). The Passu and Gulkin glaciers are located very near to the Karakoram highway connecting Pakistan with China, and there are a number of small villages (Passu, Hussaini, Gulmit, and others) close by. The Barpu and Mear glaciers are located very close to each other and around 3 km away from the residential area of the Hopar and Nagar valleys. There is a small city (Astore) near the Sachin glacier and some restaurants near its terminus (Muhammad et al., 2016). Winter snow samples were collected from mountain valleys near the Passu, Barpu, and Sachin glaciers, and three other areas to the west with a number of small villages (Figure 1). The average elevation of the selected glaciers was quite low compared to the elevation of the glaciers studied for BC, OC, and dust on the Tibetan plateau by previous researchers. The mountains around the selected glaciers are mostly dry and rocky. The 10-year records (1999–2008) of the two nearby climate stations at Khunjerab (36.83°N, 75.40°E, 4,730 masl) and Naltar (36.29°N, 74.12°E, 2,858 masl) show mean total annual precipitation values of 170 mm and 680 mm, respectively. The daily average temperature during winter and pre-monsoon showed an increasing trend between 1980 and 2014 (Gul et al., 2017). The study area is mostly exposed to the westerlies and emissions from South Asia. Most of the people in the region use wood for cooking and heating.

### 2.2 Sample collection

A total of 50 surface ice and 49 snow samples were collected from the glaciers in summer 2015 and 2016 (Passu 15, Gulkin 31, Barpu 6, Mear 8, Sachin 35, Henarche 4), and 13 in autumn 2016 (Gulkin 7, Sachin 6) at elevations ranging from 2,569 to 3,895 masl (Figure 1). Eighteen snow samples were collected in winter 2015 and 2016 from nearby mountain valleys at elevations of 1,958 to 2,698 masl; the winter sampling region was divided into six sites (S1 to S6) based on geographical location and elevation (Figure 1). Samples were collected using the "clean hands – dirty hands" principle (Fitzgerald, 1999). Ice samples were collected from the surface (5 cm depth) at different points on the glaciers. The elevation difference between collection points on the same glacier ranged from 30 to 100 meters.

The samples were preserved in ultra clean plastic bags, allowed to melt in a temporary laboratory near the sampling location, and filtered through quartz-filters immediately after melting. An electric vacuum pump was used to accelerate filtration. The melted snow/ice volume of the samples was measured using a graduated cylinder. Sampled filters were carefully packed inside petri-slides marked with a unique code representing the sample.

The snow density of winter snow samples was measured using a balance; snow/ice grain sizes were observed with an accuracy of 0.02 mm using a hand lens (25×) (Aoki et al., 2011); and snow shape was estimated using a snow card. In the models, we assumed external mixing of snow and aerosol particles and spherical snow grains.

Snow grain size and snow texture were the largest sources of uncertainty in albedo reduction (Section 3.3). Qian et
al. (2015) have summarized the sampling methods available for light absorbing particles in snow and ice from
different regions including the Arctic, Tibetan Plateau, and mid-latitude.
**2.3 Dust, OC, and BC analysis**
Before analysis, sampled filters were allowed to dry in an oven for 24 hours and then weighed using a
microbalance. The dust mass on the filters was calculated from the mass difference in weight before and after
sampling (Kaspari et al., 2014; Li et al., 2017).
There are many methods available for analyzing BC and OC. The three methods considered most effective for
measuring BC and water insoluble OC concentrations in snow are thermal optical analysis, filter-based analysis, and
single particle soot photometer analysis (Ming et al., 2008). The thermal optical (filter-based) analysis method has
been used by many researchers (e.g., Li et al., 2017) and was chosen for the study. This is an indirect method for
measuring BC and OC on sampled filters; it follows Beer's law and uses stepwise combustion of the particles
deposited on quartz filters (Boparai et al., 2008), followed by measurement of light transmission and/or reflectance
of the filters. The BC and OC content in the collected samples was measured using a thermal optical DRI carbon
analyzer, similar to the IMPROVE protocol (Cao et al., 2003). The temperature threshold applied to separate the two
species is described in Wang et al. (2012). A few (<10) filters had higher dust loads; for these the method was
slightly modified using a 100% helium atmosphere and temperature plateau (550°C). A very few (<5) samples with
very dense dust concentrations were not properly analyzed by the instrument and were excluded from the results.
The extremely high dust value of one sample from Passu (15 times the level in the next highest sample) which had
low values of other pollutants was excluded as a probable error. In some cases, a single sample was analyzed two or
three times to ensure accurate results were obtained.
The CALIPSO models also define multiple aerosol sub-types – clean continental, clean marine, dust, polluted
continental, polluted dust, smoke, and other – using the 532-nm (1064-nm) extinction-to-backscatter ratio. The
frequency of these different aerosol subtypes in the atmosphere over the study region was investigated using
CALIPSO data for the same months in which ice and snow samples were collected, i.e. January, May, June, and
December, over the period June 2006 to December 2014. The CALIPSO Level 2 lidar vertical feature mask data
product describes the vertical and horizontal distribution of clouds and aerosol layers (downloaded from
<https://eosweb.larc.nasa.gov/project/calipso/aerosol_profile_table>). The aerosol subtypes were classified in the
downloaded data using the observed backscatter strength and depolarization values. The details of the algorithm
used for classification are given in Ali H. Omar et al. (2009). The percentage contributions of individual aerosol
subtypes were plotted using MATLAB (MathWorks, Inc.).
The frequencies of different subtypes were calculated along the specific paths followed by CALIPSO over the
study region.

**2.4 Albedo simulations and estimation of radiative forcing**

Snow albedo was estimated for each of the 18 winter samples and the average calculated for samples at each of the sites (1 to 6). Albedo from two sites – S1 (Sost), which had the highest average concentration of BC and dust, and S6 (Kalam), which had the lowest average concentration of BC and dust – were further explored using the SNICAR model (Flanner et al., 2007). The aim was to quantify the effect of BC, dust, and mass absorption cross-section (MAC) on albedo reduction. Sensitivity model experiments were carried out using various combinations of BC, dust, and MAC values, while other parameters were kept constant (parameters for sites 1 and 6 shown in supplementary materials, Table S1). Snow albedo was simulated for different daylight times, with the SZA set in the range 57.0–88.9° based on the position of the sun in the sky for the sampling date and locations. The daily mean was calculated from the mean of the albedo values simulated for 24 different SZA values (one per hour), and the daytime mean from the mean of the albedo values simulated for 10 SZA values (one per hour during daylight). The mid-latitude winter clear-sky option was selected for surface spectral distribution. The parameters used for sensitivity analysis are shown in Table S1. MAC values of 7.5, 11, and 15 $m^2/g$ were selected based on a literature review (Que et al., 2014; Pandolfi et al., 2014). In order to reduce the uncertainty, the dust concentration in the samples was divided into four diameter classes (as per the model requirements): size 1 (0.1–1.0 µm) was taken to be 2%, size 2 (1–2.5 µm) to be 13%, size 3 (2.5–5 µm) to be 31%, and size 4 (5–10 µm) to be 54% of total dust mass present in the sample, based on results published by others (Gillette et al., 1974; Mahowald et al., 2014). Radiative forcing (RF) was estimated for the same samples following Eq. (1):

$$RF_x = R_{in-short} * \Delta \boldsymbol{\alpha}_x \qquad\qquad (1)$$

where $R_{in-short}$ denotes incident short-wave solar radiation (daily mean), as measured by the SBDART (Santa Barbara DISORT Atmospheric Radiative Transfer) model, and $\Delta\alpha_x$ denotes the daily mean reduction in albedo, as simulated by the SNICAR model.

**2.5 Source regions of pollutants**

Three methods were used to identify the potential source regions of pollutants found at the study site: wind maps, emissions inventory coupled with back trajectories, and a region-tagged chemical transport modeling analysis.

Wind vector maps were prepared using MERRA-2 reanalysis data (available from the National Aeronautics and Space Administration [NASA] https://gmao.gsfc.nasa.gov/reanalysis/MERRA-2/docs/). The U and V wind components were combined into a matrix around the study area for each individual month and then plotted against latitude/longitude values to show the spatial variance of monthly wind stress at 850 mb using arrows to indicate the direction and intensity of wind.

Air trajectories were calculated backwards from the sampling sites (S1: 36.40°N 74.50°E; S6: 35.46°N 72.54°E) to identify potential source regions for the pollutants using the web version of the Hybrid Single Particle Lagrangian Integrated Trajectory (HYSPLIT-4) model (Draxler and Hess, 1998). The HYSPLIT-4 model has been used by others to compute air mass trajectories to identify possible source regions (Ming et al., 2009; Zhang et al., 2013). Reanalysis

meteorological data from the same source as the wind data (https://www.esrl.noaa.gov/psd/data) were used as input
data in the HYSPLIT model for May, June, and December 2015, and January 2016. HYSPLIT was run in a seven-day
backward trajectory mode with trajectories initiating every six hours (0, 6, 12, and 18) on a daily basis from 4 May to
19 June 2015 (77 days during summer) and from 1 December 2015 to 31 January 2016 (62 during winter). The
HYSPLIT model results were combined with Representative Concentration Pathways (RCPs) emission data for 2010
(available from http://sedac.ipcc-data.org/ddc/ar5_scenario_process/RCPs.html; data file <RCPs_anthro_BC_2005-
2100_95371.nc>) to identify the source location. This comprises emission pathways starting from an identical base
year (2000) for multiple pollutants, including BC and OC; the file description indicates that the inventory includes
biomass burning sources. The RCP inventory has the same emissions sectors as the Hemispheric Transport Air
Pollution (HTAP) emission inventory used in the modelling approach for identifying source regions (see below),
including fuel combustion, industry, agriculture, and livestock, but the HTAP inventory has a higher resolution (0.1 x
0.1 degree) than the RCP inventory (0.5 x 0.5 degree). Lamarque et al. (2010) give a more detailed discussion of the
inventory and sectors (12) used in the base year calibration of the RCP. Monthly CALIPSO satellite-based extinction
data from 2006 to 2014 were used to calculate the vertical profile for aerosol extinction over the study region. The
CALIPSO extinction profile was constructed for selected months in 2006 to 2014 – May and June for summer and
December and January for winter (Figure S1). The exponential equation $X = (\log(10.46) - \log(Y))/10.29$ , where
$Y$ is the vertical height of individual trajectories in kilometers and $X$ indicates the extinction against the height of
trajectories, was used to calculate the extinction profile for the trajectory heights. The normalized extinction profile
was obtained by assuming that surface extinction = 1 (Figure S1).

The WRF-STEM model was used as a third approach for identifying the origin (source regions) of air masses

carrying pollutants. The WRF-STEM model uses region-tagged carbon monoxide (CO) tracers for many regions in
the world to identify geographical areas contributing to observed pollutants (Adhikary et al., 2010). Region tagged
CO tracers are used as a standard air quality modeling tool in various regional and global chemical transport models
to identify pollution source regions (Chen et al., 2009; Park et al., 2009; Lamarque and Hess, 2003). The WRF-STEM
model domain was centered on 50.377° E longitude and 29.917° N latitude, with a model horizontal grid resolution
of 45 x 45 km with 200 grids in the east-west direction and 125 north-south. The meteorological variables needed for
the chemical transport were derived from the Weather Research and Forecast (WRF) meteorological model (Grell et
al., 2005) using FNL data (ds083.2) available from the UCAR website as input data. The main aim of the simulation
was to identify the geographic locations contributing to the observed pollutants at the field sites. The HTAP version 2
emission inventory, which comprises multiple pollutants including BC and OC, was used for the WRF-STEM
modeling (available from <http://edgar.jrc.ec.europa.eu/htap_v2/>). This emission inventory includes major sectors
such as energy, industry, transport, and residential, but not large scale open agricultural and open forest burning. The
simulations applied in our study used the anthropogenic emissions from the HTAP inventory. Thus the results indicate
the amount of pollutants reaching the study area from day-to-day planned and recurring activities in domestic,
transport, industrial, and other sectors. The model was run for a month prior to the field campaign dates to allow for
model spin up (normal practice for a regional chemical transport model), and then for the months of December, January,
and June, to match the field campaign dates.
**3. Results and discussion**
**3.1 BC, OC and dust concentrations**
The minimum, maximum, and average concentrations of BC, OC (water insoluble organic carbon), and dust in
the ice and snow samples are given in Table 1. The OC, and BC concentration values were blank corrected by
subtracting the average value of the field blanks. Blank concentrations were used to calculate detection limits as
mean ± standard deviation. The average BC concentration overall was $2130 \pm 1560$ ng g$^{-1}$ in summer samples, 2883
$\pm 3439$ ng g$^{-1}$ in autumn samples (both from glaciers), and $992 \pm 883$ ng g$^{-1}$ in winter samples. The average OC
concentration overall was $1839 \pm 1108$ ng g$^{-1}$ in summer samples, $1423 \pm 208$ ng g$^{-1}$ in autumn samples, and $1342 \pm$
672 ng g$^{-1}$ in winter samples. There was considerable variation in individual samples, with summer values of BC
ranging from 82 ng g$^{-1}$ (Gulkin glacier) to 10,502 ng g$^{-1}$ (Henarche glacier), autumn values from 125 ng g$^{-1}$ (Gulkin
glacier) to 6481 ng g$^{-1}$ (Sachin glacier), and winter samples from 79 ng g$^{-1}$ (Kalam) to 5957 ng g$^{-1}$ (Sost).
The lowest BC (82 ng g$^{-1}$) and OC (128 ng g$^{-1}$) concentrations were observed in summer samples collected from
the Gulkin and Sachin glaciers, respectively. The average values of BC and OC were low in all samples from the
Passu glacier, even though it lies close to the Karakoram highway which links Pakistan with China. The low
concentrations of BC may have been due to the east facing aspect of the glacier shielding it from pollutants
transported from west to east. Slope aspect of a glacier is known to be important for snow cover dynamics (Gul et
al., 2017); dust concentrations are known to vary with slope aspect due to the effects of wind direction on
deposition.
The highest average concentration of BC was found in autumn samples from the Sachin glacier, and the highest
average concentration of OC in summer samples from the same glacier. The average concentration of BC was much
greater in autumn than in summer on the Sachin glacier, but somewhat greater in summer than in autumn on the
Gulkin glacier, indicating highly spatiotemporal patterns in the deposition of particles. The marked difference on the
Sachin glacier may have reflected the difference in the direction of air, which comes from Iran and Afghanistan in
summer and the Bay of Bengal via India in autumn, with the generally lower deposition on the Gulkin glacier more
affected by other factors (such as slope aspect of the glacier and status of local emissions near the glacier). There
was no clear correlation between the average BC concentration in glacier samples and glacier elevation. However,
winter snow samples showed a weak increasing trend in average BC with site elevation (Table 1, Figure S3).
Most summer samples were collected from surface ice (Figure S2 a), but a few samples for Gulkin and Sachin
were collected from aged snow on the glacier surface (Figure S2 b,c). Dust was visible on the relatively aged snow,
and the BC and OC concentrations in these snow samples were much higher than those in ice. The highest average
BC values in winter were also observed in aged snow (from Sost) and the lowest in fresh snow (from Kalam) (Table
1). Generally, snow samples collected within 24 hours of a snowfall event were considered as a fresh snow.
We analyzed the ratios of OC to BC in the different samples as in atmospheric fractions this can be used as an
indicator of the emission source, although apportionment is not simple and only indicative. The BC fraction is
emitted during combustion of fossil fuels, especially biomass burning in rural areas in winter, and urban emissions
from road transport. The OC fraction can be directly emitted to the atmosphere as particulate matter (primary OC)
from fossil fuel emissions, biomass burning, or in the form of biological particles or plant debris; it can also be
generated in the atmosphere as gases are converted to particles (secondary OC). In general, lower OC/BC ratios are
associated with fossil fuel emissions and higher OC/BC ratios with biomass burning. Overall, there was no clear
correlation between BC and OC concentrations in our samples. In most cases, the concentration of OC was greater
than the concentration of BC, which might indicate a greater contribution from biomass burning in the emissions,
but in a few the concentration of BC was greater than that of OC, which might indicate a greater contribution from
coal combustion. The lowest OC/BC ratio of 0.041 was observed in a summer sample from Henarche glacier, and
the highest ratio of 5 in a winter sample from Kalam. The higher value at Kalam may indicate greater contributions
from biomass burning than from fossil fuel combustion in the region. In summer samples, the average concentration
of OC was greater than the average concentration of BC in samples from four of the six glaciers, but it was much
lower in Barpu and Henarche. In winter, individual snow samples indicated that concentration of OC was greater
than BC at low elevation sites and vice versa; the average OC was greater than average BC at all except the highest
elevation site (Table 1).
However, these results should be considered with care. There are a number of factors that can affect the OC/BC
ratio in snow and ice samples apart from the concentrations in the atmosphere. Spatio-temporal variability of the
OC/BC ratio may indicate the contribution of different sources, seasonal variation, and frequent change in wind
direction. In deposited samples, low OC/BC ratios can result from a reduction in OC (Niu et al., 2017), greater
contributions from BC enrichment and OC scavenging, and/or the contribution of different emission sectors
(including quantity, combustion conditions, and fuel type). Post deposition processes of scavenging and enrichment,
which are influenced by snow melt rate, can cause water soluble OC to be under-represented as meltwater removes
OC but not BC, with OC and BC being redistributed primarily by meltwater rather than by sublimation and/or
dry/wet deposition. Thus the OC/BC ratio often reflects the impact of dilution of dissolved organic carbon and
enrichment of primary organic carbon during snow/ice melting, with differences in OC/BC ratios reflecting
differences in the enrichment process. The low OC/BC ratio in the samples from Henarche, the glacier at the lowest
elevation, could, for example, be due to preferential washing out of OC particles with meltwater. Overall, there was
a higher positive correlation between BC and dust than with OC, suggesting that for BC and dust, particle
precipitation and enrichment processes were similar. The method used for analysis and the amount of dust loading
on the sample can also affect the OC/BC ratio, as can the presence of metal oxides and calcium carbonate. High iron
oxide concentrations can cause BC to pre-oxidize or drop off the filter, while calcium carbonate can be wrongly
identified as BC. Laboratory studies have shown that the presence of metal oxides in aerosol samples can alter the
OC/BC ratio either by enhancing OC charring or by lowering the BC oxidation temperature (Wang et al., 2010),
while higher fractions of metal oxide can increase BC divergence across the thermal optical protocols (Wu et al.,
2016). Dust can lead to a greater decrease in optical reflectance during the 250°C heating stage in the thermal/optical
method, and thus an incorrect OC/BC ratio (Wang et al., 2012). Carbon detected by the flame ionization detector
(FID) before the optical signal attains the initial value is defined as OC and that detected after is defined as BC; dust
on the filter results in the FID division being postponed or inefficient, and thus OC being overestimated and BC
underestimated or even negative (Wang et al., 2012). Wang et al. (2012) provides a more detailed discussion of
OC/BC ratios derived using the thermal optical method.

A wide range of values has been reported by different authors for BC concentrations in snow and ice samples from

different regions (Table S2). The concentrations of BC in our samples were higher than those reported by many authors
(Table S2), but were comparable with the results reported by Xu et al. (2012) in the Tien Shan Mountains, Li et al.
(2016) in the northeast of the Tibetan plateau, Wang et al. (2016) in northern China, and Zhang et al. (2017) in western
Tien Shan, Central Asia. High concentrations indicate high deposition rates on the snow and ice surface, but there are
several possible reasons for a wide variation in values apart from differences in deposition rates, including differences
in sampling protocols, geographical/sampling location and elevation of sampling site (Qu et al., 2014), and year/season
of sampling. The majority of samples were from the ablation zone of the glaciers. Strong melting of surface snow and
ice in the ablation zone could lead to BC enrichment and high BC concentrations, as observed by Li et al. (2017) for
glaciers on the Southern Tibetan Plateau. The sampling season (May to September in our study) is an important factor
because rapid enrichment occurs as snow melts during the melting season. The peak melting period is May to
August/September, thus the concentration of BC, OC, and dust in our samples would have been increased as melting
progressed due to the enrichment in melting snow and scavenging by the melting water. In most cases, snow and ice
samples were collected quite a long time after snow fall, and the concentration of pollutants would also have increased
in the surface snow and ice due to dry deposition. It seems likely that the pollutants in surface samples would be
affected by sublimation and deposition until the next melt season (Yang et al., 2015). In some of the cases in our study,
the average concentration of BC, OC, and/or dust for a particular glacier/site was increased as a result of a single
highly concentrated sample, reflecting the wide variation that results from the interplay of many factors.

Enrichment is more marked at lower elevations as the temperatures are higher which enhances melting and

ageing of surface snow, while deposition also tends to be higher because the pollutant concentrations in the air are
higher (Wang et al., 2012; Nair et al., 2013). Previous studies have tended to focus on the accumulation area of
glaciers (e.g. ice cores and snow pits) where enrichment influences are less marked, and on high elevation areas,
where deposition is expected to be lower, in both cases leading to lower values. In our study, the majority of samples
collected in summer and autumn were collected from the ablation area of debris-covered glaciers where enrichment
influences are marked due to the relatively high temperature, and this is reflected in the relatively high values of BC,
OC, and dust. Li et al. (2017) showed a strong negative relationship between the elevation of glacier sampling
locations and the concentration of light absorbing particles. Stronger melt at lower elevations leads to higher
pollutant concentrations in the exposed snow. Equally, BC may be enriched in the lower elevation areas of glaciers
as a result of the proximity to source areas, as well as by the higher temperatures causing greater melting. Thus the
main reason for the high concentrations of BC, OC, and dust in our samples may have been that the samples were
taken from relatively low elevation sites. Human activities near the sampling sites in association with the summer
pilgrimage season probably also contributed to an increase in pollutant concentrations. Our results do not necessarily
indicate that all the glaciers in the Karakoram region are substantially darkened by BC. The ablation zones of debris
covered glaciers which are at relatively low elevations and near to pollution sources may be more polluted than
other glacier areas.

**3.2 Frequency distribution of aerosol sub types in the atmosphere**

The analysis of aerosol types using the CALIPSO data identified smoke as the most frequent aerosol type over
the study region in both summer and winter, indicating that biomass burning may be the dominant source of
emissions. Figure 2 shows the average frequency of different aerosol types in May/June (summer) and
December/January (winter) over the period 2006 to 2014 in the form of a box plot. The frequency of different
aerosol subtypes in June from 2006 to 2014 is shown in Figure S4; smoke had the highest frequency (39%),
followed by dust (21%), polluted dust (12%), and other (20%). This type of aerosol measurement in the atmosphere
was useful for our current study because it provides observation based data over the study region, whereas the other
approaches used (such as modeling) were based on interpolation not observation. Pollutant deposition depends on
the concentration of pollutants in the atmosphere and the results are consistent with the high concentration of BC
(from smoke) and dust particles in the glacier and snow surface samples.

**3.3 Snow albedo reduction**

The albedo of individual winter snow samples was calculated using the SNICAR model and then averaged for
each site (S1 to S6). Figure 3a shows the average for each site across the visible and infrared spectrum. Two sites
were chosen for further analysis: S1 (Sost) which had the highest average concentration of BC, and S6 (Kalam)
which had the lowest average concentration of BC. The albedo was simulated for different MAC values and SZA for
samples at the two sites as described in the methods. The values for average albedo of samples from the two sites
simulated for MAC values of 7.5, 11, and 15 m$^2$/g and SZA of 57.0–88.9° (daytime) under a clear sky ranged from
0.39 (site S1, BC only, midday, MAC 15 m$^2$/g) to 0.85 (site S6, dust only, early evening, MAC 7.5–15 m$^2$/g). The
detailed values are shown in Table S3.
The percentage change in albedo was calculated in absolute terms as the change between albedo values with a
pollutant (BC or dust or both) and a reference albedo value with zero pollutants (zero BC and dust concentration).
Table 2 shows the calculated percentage reductions in daily minimum, maximum, and mean broadband snow albedo
at different MAC values (7.5, 11, 15 m$^2$/g) resulting from the average BC, dust, and combined BC and dust
concentrations found in samples at each of the sites. The reduction was strongly dependent on BC concentration and
almost independent of dust concentration, and increased with increasing MAC value. The results suggest that BC
was the dominant forcing factor, rather than dust, influencing glacial surface albedo and accelerating glacier melt.
BC was found to play an important role in forcing in the northern Tibetan plateau (Li et al., 2016), whereas in the
central Tibetan plateau and Himalayas, dust played a more important role (Qu et al., 2014; Kaspari et al., 2014). The
MAC value affected the albedo more in the visible range than at 1.2 μm (near infrared) wavelength (Fig 3c,d). The
combined concentration of BC and dust, or BC alone, strongly reduced the snow albedo for a given combination of
other input parameters. The effect at the low pollutant site (S6) was small: the values for daytime snow albedo at
0.975 μm due to BC, or BC plus dust with different MAC and SZA, ranged from 0.70 to 0.83, with a reduction in
daily mean albedo of 1.8 to 2.9%, and those for dust alone from 0.79 to 0.85, with a reduction in daily mean albedo
of less than 0.1%. The effect at the high pollutant site (S1) was much more marked: BC or BC and dust reduced
daytime snow albedo to values ranging from 0.39 to 0.64, a reduction in daily mean albedo of 8.8 to 12.0%, but the
effect of dust alone was still low with values of 0.70 to 0.78, again a reduction in daily mean albedo of less than

0.1%.

Both the snow albedo and the impact of light absorbing particles depend on a range of factors including the SZA,

snow depth, snow grain size, and snow density. For example, the snow albedo reduction due to BC is known to be
less in the presence of other light absorbing particles as these will absorb some of the available solar radiation
(Kaspari et al., 2011). The snow albedo calculated for our samples was strongly dependent on the SZA with albedo
increasing with decreasing SZA, especially at near infrared wavelengths (Table S3).

The impact of snow ageing was also investigated. The winter samples from S1 (Sost) were aged snow, whereas

those from S6 (Kalam) were fresh snow (Table 1, Figure S5 b,c). Not only was dust clearly visible on the surface of
the aged snow, the grain size was large and the snow was dense. The aged snow had a much higher concentration of
BC and dust, which reduced the albedo, but the extent of reduction is also affected by other factors. Albedo
reduction by BC and dust particles is known to be greater for aged snow than for fresh snow (Warren and
Wiscombe, 1985). In our samples, the calculated reduction in snow albedo for high MAC values (15) compared to
low MAC values (7.5) was greater in aged snow than in fresh snow (Figure 3b). The effective grain size of snow
increases with time as water surrounds the grains. Snow with larger grain size absorbs more radiation because the
light can penetrate deeper into the snowpack, thus decreasing surface albedo (Flanner et al., 2006). In the melting
season, the snowpack becomes optically thin and more particles are concentrated near the surface layer, which
further increases the effect on albedo.

The estimated reduction in snow albedo by dust and BC (up to 29% of daytime maximum value, Table 2) was

higher than that reported by others for High Asia based on farmers' recordings (e.g 1.5 to 4.6% reported by Nair et
al., 2013) and in the Himalayas (Ming et al., 2008; Kaspari et al., 2014; Gertler et al., 2016). However, although the
values were relatively high, they were at the same level or lower than the estimates for albedo reduction of 28% by
BC and 56% by dust in clean ice samples, and of 36% by BC and 29% by dust in aged snow samples, reported by
Qu et al. (2014) for surface samples from the Zhadang glacier, China. Simulation results by Ming et al. (2013a)
showed BC, dust, and grain growth to reduce broadband albedo by 11%, 28%, and 61%, respectively, in a snowpack
in central Tibet. Dust was the most significant contributor to albedo reduction when mixed inside the snow and ice,
or when the glacier was covered in bare ice. In our case BC was a more influential factor than dust during a similar
study period to that reported by Li et al. (2017), indicating that BC plays a major role in albedo reduction.
The possible reasons for the relatively high values for albedo reduction in our samples include the lower
elevation of the sampling locations, relatively high concentrations of BC and dust, high MAC values, low snow
thickness, underlying ground quality, presence of small and large towns near the sampling sites, and predominance
of aged snow samples. Most of the samples collected in winter were from places with a snow depth less than 50 cm
(Figure S5a), thus mud, stones, and clay below the snow layer would be expected to increase the absorption of solar
radiation and reduce the albedo.
The high albedo reduction in the visible range of the electromagnetic spectrum could be due to the relatively
high concentration of surface (~1cm) snow impurities. The total amount of deposited particles in the surface layer of
aged snow was relatively high, indicating a high deposition rate of atmospheric pollutants.
Flanner et al. (2007) reported that BC emission and snow ageing are the two largest sources of uncertainty in
albedo estimates. The uncertainties in our estimated albedo reduction include the BC type (uncoated or sulfate coated),
the size distribution of dust concentration, the accuracy of snow grain size, snow texture, snow density, and albedo of
the underlying ground. Sulfate-coated particles have an absorbing sulfate shell surrounding the carbon; recent studies
confirm that coated BC has a larger absorbing power than non-coated BC (Naoe et al., 2009). We used uncoated black
carbon concentration in the SNICAR model, but the pollutants at the remote site are presumed to be mainly from long
range transport, thus the BC may have gained some coating. The albedo reduction for sulfate-coated black carbon was
calculated to be 3–8.5% higher, depending on the MAC and SZA values, than for uncoated black carbon at the low
concentration site S6 (Figure S6). The snow grain size (snow aging) and snow texture are also large sources of
uncertainty. The effect of snow grain size is generally larger than the uncertainty in light absorbing particles and varies
with the snow type (Schmale et al., 2017). The albedo reduction caused by 100 ng g$^{-1}$ of BC for an effective snow
grain radius of 80 μm, 100 μm, or 120 μm was calculated to be 0.017, 0.019, or 0.021, respectively. The snow grain
size was measured with a hand lens with an accuracy of 20 μm, thus the associated uncertainty in the albedo results
was at least 0.002. The snow grain shape was measured with the help of a snow card, but a spherical shape was
assumed for snow grains in the (online) SNICAR albedo simulation model. The albedo of non-spherical grains is
higher than the albedo of spherical grains (Dang et al., 2016), and the shape of snow grains and/or ice crystals changes
significantly with snow age and meteorological conditions during and after snowfall (LaChapelle 1969). A number of
recent studies (e.g., Flanner et al., 2012; Liou et al., 2014; He et al., 2014, 2017) have shown that both snow grain
shape and aerosol-snow internal mixing play an important role in snow albedo calculations.
**3.4 Radiative forcing (RF)**
Radiative forcing (RF) is a measure of the capacity of a forcing agent to affect the energy balance in the

atmosphere – the difference between sunlight absorbed by the Earth and energy radiated back to space – thereby contributing to climate change. Changes in albedo contribute directly to radiative forcing: a decrease in albedo means that more radiation will be absorbed and the temperature will rise. In snow and ice, the additional energy absorbed by any pollutants present also increases and accelerates the melting rate.

Various authors have described the impact of albedo change in snow and ice on radiative forcing. Zhang et al. (2017) reported that a reduction in albedo by 9% to 64% can increase the instantaneous radiative forcing by as much as 24.05–323.18 $Wm^{-2}$. Nair et al. (2013) estimated that in aged snow a BC concentration of 10–200 $ng\ g^{-1}$ can increase radiative forcing by 2.6 to 28.1 $Wm^{-2}$; while Yang et al. (2015) reported radiative forcing of 18–21 $Wm^{-2}$ for aged snow in samples from the westernmost Tibetan Plateau. The authors used different atmospheric conditions in the forcing estimates: Zhang et al. (2017) used mid-latitude winter with clear sky and a cloudy environment, Nair et al. (2013) mid-latitude winter, and Yang et al. (2015) clear-sky and cloudy conditions .

We calculated the radiative forcing in the samples assessed for daytime albedo and daily (24h) mean albedo. The radiative forcing at different daylight times caused by BC deposition varied from 3.93 to 43.44 $Wm^{-2}$ (3.93–11.54 $Wm^{-2}$ at the low BC site and 20.88–43.45 $Wm^{-2}$ at the high BC site), and that by dust from 0.16 to 2.08 $Wm^{-2}$ (0.16–0.30 $Wm^{-2}$ at the low BC site and 1.38–2.08 $Wm^{-2}$ at the high BC site) (detailed values given in Table S4), indicating that BC was the dominant factor. The radiative forcing due to combined BC and dust was very similar to that for BC alone. In contrast, studies by others have shown higher forcing by dust than by BC based on the optical properties and size distribution of dust particles (Qu et al., 2014). In our study, the increase in daily mean radiative forcing ranged from 0.1% for dust only at the low pollutant site to 14.9% for BC at the high pollutant site. However, dust forcing varies strongly with dust optical properties, source material, and particle size distribution. The properties for dust are unique in each of the four size bins used in the SNICAR online model. These size bins represent partitions of a lognormal size distribution. We used an estimated size of dust particles and generic dust properties in the model, but some dust particles can have a larger impact on snow albedo than the dust applied here (e.g., Painter et al., 2007).

Both radiative forcing and albedo reduction increased with decreasing daytime SZA, indicating higher melting at midday compared to morning and evening. Figure 4 shows the daily mean albedo reduction and corresponding radiative forcing caused by BC for fresh (low BC) and aged (high BC) snow with different MAC values. Snow aging (snow grain size) plays an important role in albedo reduction and radiative forcing. According to Schmale et al. (2017) the effect of snow grain size is generally larger than the uncertainty in light absorbing particles, which varies with snow type. Snow aging reduces snow albedo and accelerates snow melt, but the impact of snow aging on BC in snow and the induced forcing is complex and includes spatial and seasonal variation (Qian et al., 2014).

An increase in MAC value from 7.5 to 15 led to an increase in radiative forcing by 1.48 $Wm^{-2}$ in fresh snow and 4.04 $Wm^{-2}$ in aged snow. This suggests that when the surface of snow, ice, and glaciers experience strong melting, enrichment with BC and dust could cause more forcing. Previous studies of ice cores and snow pits probably underestimated the albedo reduction and radiative forcing in glacier regions as samples were taken from high

elevation areas where there is less ageing and melting and thus lower surface enrichment of BC and dust than at
lower elevation. Our results are higher than those reported in other studies on the northern slope of the Himalayas
(Ming et al., 2012), western Tibetan Plateau (Yang et al., 2015b), and Tien Shan mountains (Ming et al., 2016).
However, they are comparable to values for radiative forcing reported more recently by others, for example for the
Muji glacier (Yang et al., 2015), Zhadang glacier (Qu et al., 2014), in high Asia (Flanner et al., 2007; Nair et al.,
2013), and in the Arctic (Wang et al., 2011; Flanner, 2013). The results suggest that enrichment of black carbon (in
our case) and mineral dust (other authors) can lead to increased absorption of solar radiation, exerting a stronger
effect on climate and accelerating glacier melt.
**3.5 Potential source regions**
**3.5.1 Wind vector maps**
Figure 5 shows the spatial variance of wind vector maps (U and V) at 850 hPa in May, June, January, and
December prepared using MERRA-2 reanalysis data for the year 2015/2016. The wind blows primarily from west to
east but there were variations over the year. Central Asia contributed some part of the air in May and June. In May,
the prevailing air masses were from Syria, Turkey, Turkmenistan, Iraq, Azerbaijan, northwest Iran, Afghanistan,
Nepal, southwest China, and southern Pakistan; the trend was similar in June with but with a smaller contribution
from Nepal and southwest China. In December and January (winter), the western trade winds were stronger than the
easterlies and the wind blew from Azerbaijan and northwest Iran, reaching the study site via Syria, Iraq,
Turkmenistan, and Afghanistan.
**3.5.2 Coupled emissions inventory with back air trajectory**
Trajectory analysis using the HYSPLIT model showed that in May and June 2015 air parcels reached the study
site along three different pathways: one from north Asia (Russia) via Central Asia (Kazakhstan), one from western
Asia (Cyprus and Syria) via Central and Southern Asia (Georgia), and one via India, which was more local (Figure
6). The trajectories in summer had distinct pathways, while those in winter were dispersed in all directions, partially
covering West, East, and South Asia, and completely covering Central Asia. Figure 6 shows the product of
extinction and emission calculated along the pathways of trajectories calculated using the vertical profile for aerosol
extinction over the study region obtained from the monthly CALIPSO satellite-based extinction data. Scattering and
absorption decreased exponentially with increasing elevation (Figure S1) but was still visible at elevations above 5
km in summer.
The RCP emission data combined with back trajectories and extinction data showed that the hotspot regions of
pollution that affected the study sites during winter were mainly to the southwest rather than very distant (Figure
6b). Iran, Turkmenistan, Azerbaijan, Georgia, the eastern part of Turkey, and the southwestern part of Russia all
showed comparatively high pollutant emissions in winter which moved towards northern Pakistan. The western part
of Kazakhstan, Uzbekistan, and northeastern Turkey emitted particularly high concentrations of pollutants.
Combination of the back-trajectory results and surface-wind direction analysis indicated that during the sampling
months, aerosols were significantly influenced by the long-range transport of pollutants coming from Central and
South Asia, with a small contribution from West and East Asia. This differs somewhat from previous reports which
suggested that the Tibetan Plateau and Himalayan region are mainly affected by pollutants from East and South Asia
(Zhang et al., 2015). An increasing trend has been reported for black carbon emissions in Central and South Asia
over the past 150 years (Bond et al., 2007), and a significant increase has been found in black carbon concentrations
in glacier snow in west China in the last 20 years, especially during the summer and monsoon seasons (Ming et al.,
2008). In South Asia, the largest source of atmospheric black carbon is emission from biomass and biofuels used for
cooking and heating (dung, crop residues, wood) (Venkataraman et al., 2005).
The results indicate that only a low level of pollutants (minor contribution) reached the study area from
northwest China. BC particles emitted from distant low latitude source regions such as tropical Africa barely reach
the Tibetan Plateau and Himalayan regions because their emissions are removed along the transport pathways during
the summer monsoon season (Zhang et al., 2015).
**3.5.3 Chemical transport modelling**
The contribution of pollutants from potential source regions was also investigated using the WRF-STEM model
with tagged carbon monoxide tracers and source regions of East Asia, South Asia, Central Asia, the Middle East,
Europe, the Russian Federation, and West Asia. (The individual countries in the regions are listed in Table S5).
Figure 7 shows the results of the model simulations for summer (1 June to 4 July 2015) and winter (15
December 2015 to 17 January 2016) at two glacier sites (Sachin and Shangla) where the model terrain elevation was
close to the observation terrain elevation. The model simulations showed Pakistan to be the major contributor of
pollutants in summer (77% at Shangla and 43% at Sachin) followed by the South Asian countries; and the south
Asian countries in winter (47% at Shangla and 71% at Sachin) followed by Pakistan, which is in line with the
findings by Lu et al. (2012) that South Asia contributed 67% black carbon in the Himalayas. There were minor
contributions of 2–7% of pollutants from Afghanistan, Iran, Central Asia, and the Middle East, and extremely small
amounts from East Asia, Europe, Africa, West Asia, and China. The contribution from Iran, the Middle East, and
Europe was greater in winter than in summer, while the contribution from Central Asia and China was greater in
summer than in winter. The proportion of daily contributions fluctuated considerably: with higher contributions from
Iran, the Middle East, and Europe on individual days in winter, ranging for example from 2–30% for the Middle
East.
The concentration of hydrophobic BC (BC1), hydrophilic BC (BC2), and total black carbon (BC = BC1 + BC2)
given by the model for the Sachin glacier grid point in the summer and winter seasons is shown in the supplemental
material (Figure S7). In the model, freshly emitted BC particles are hydrophobic and gradually acquire a
hygroscopic coating over time. A time series analysis of BC1 and BC2 concentrations shows the influence of both
freshly emitted BC and aged BC reaching the observation location. The highest concentration of BC1 was observed
on 20[th] December 2015 and the second highest on 25[th] June 2015, indicating an influence from freshly emitted air
masses in both the summer and winter months. Future studies (BC tracer) will evaluate the details of the different
source regions of the BC reaching the glaciers compared to region-tagged CO tracers.
**3.5.4 Comparison of the different approaches used to identify potential source regions**
The high BC concentration in the atmosphere over the study region was attributed to long-range transport from
urban source regions. Potential source regions of the pollutants deposited on glaciers and snow were identified using
wind vector mapping with MERRA-2 reanalyzed data, calculation of back air trajectories using the HYSPLIT-4
model, and chemical transport pathways using the WRF-STEM tagged chemical transport model. The back
trajectory results indicated that the majority of pollutants in summer were from Central and South Asia, and in
winter from Iran, Pakistan, Iraq, Turkmenistan, Azerbaijan, Georgia, Jordan, Syria, Tunisia, Ukraine, Libya and
Egypt. The WRF-STEM model indicated that most anthropogenic pollutants were from Pakistan and South Asia
during both summer and winter. However, both approaches showed a reasonable contribution from Central Asian
countries and limited contribution from East Asian countries in summer. The wind vector maps also indicated that
the study site was mostly affected by westerly winds. All three approaches showed a reasonable contribution from
neighboring countries such as Afghanistan, Pakistan, Iran, and India in specific months. Overall, the results indicate
that South, Central, and West Asia were the major sources of the pollutants detected at the sampling sites.
There was some mismatching in source regions among the three approaches. The WRF-STEM model and wind
vector maps both identified a small contribution from East Asia, but this was not identified in the back trajectories
approach. Similarly, the wind vector maps and back air trajectories showed a dominant contribution from the west,
while the WRF-STEM model showed a major contribution from Pakistan and South Asia. The differences in the
results obtained by the different methods may be due in part to the complex topography of the region and the
different altitudes used by the methods; the coarse resolution of the WRF-STEM model; and differences in the
emission source inventories and meteorological parameters used by the WRF-STEM and HYSPLIT-4 models. The
limitations of using back trajectories to identify source region is discussed further in a paper by Jaffe et al. (1999).
Furthermore, the atmospheric BC concentration over the Himalayas has significant temporal variations
associated with synoptic and meso scale changes in the advection pattern (Babu et al., 2011) which can affect
pollutant transport and deposition. The large uncertainty among different emission inventories can also affect the
results, especially in the Himalayan region.
**4 Summary and conclusion**
Black carbon (BC) and organic carbon (OC) concentrations were measured using thermal optical analysis of
snow and ice surface samples collected from glacier and mountain valleys in northern Pakistan in summer, autumn,
and winter. The samples contained high concentrations of BC, OC, and dust in low elevation glaciers and surface
snow in mountain valleys. The samples from Sost contained the highest average concentration of BC in mountain
valley snow (winter) and those from Kalam the lowest, probably due to the impact of snow age and increased
concentration of black carbon and dust (the Sost samples were aged snow and Kalam samples fresh snow). The
average concentration of BC in surface samples from the Sachin glacier was higher in autumn than in summer; the
BC values in summer snow samples collected from the Sachin and Gulkin glaciers (aged snow from the glacier
surface) were much higher than those in ice. The average BC concentration in summer samples collected from
glaciers was $2130 \pm 1560$ ng g$^{-1}$ and that in autumn samples $2883 \pm 3439$ng g$^{-1}$. The average concentration of OC
was $1839 \pm 1108$ ng g$^{-1}$ in summer samples, $1423 \pm 208$ ng g$^{-1}$ in autumn samples, and $1342 \pm 672$ ng g$^{-1}$ in winter
samples, with the highest variability in summer samples. The individual lowest BC (82 ng g$^{-1}$) and OC (129 ng g$^{-1}$)
concentrations were observed in summer samples collected from the Gulkin and Sachin glaciers, respectively. Dust
and other pollutants were clearly visible on aged snow and ice surfaces; the results indicate considerable enrichment
during ageing. The pollutant concentrations in our samples were relatively higher than those reported by others in
earlier studies, which tended to focus on the accumulation area of glaciers (e.g. ice cores and snow pits), where
enrichment influences are less marked and measured values are likely to be lower, and high elevation areas, where
deposition of pollutants is expected to be lower. It is likely that pollutant concentrations were underestimated in
these earlier studies, particularly when there was strong surface melting.
Snow albedo was calculated for winter samples using the SNICAR model with various combinations of BC and
dust concentrations, three values for MAC, and a range of values for SZA (57–88.89º during daytime), with other
parameters kept constant. BC was the major component responsible for albedo reduction, dust had little effect. The
reduction by BC ranged from 2.8 to 32.5% during daytime, which is quite high, with albedo reduced to below 0.6.
The reduction was greater for higher concentrations of BC and greater MAC. The reduction in 24 h average albedo
ranged from <0.07–2.9% for fresh snow samples and <0.05–12.0% for aged snow. Changes in albedo contribute
directly to radiative forcing: a decrease in albedo means that more radiation will be absorbed and the temperature
will rise. The radiative forcing by BC was also higher than that caused by dust, indicating that BC was the dominant
factor. The daytime albedo values in winter snow samples ranged from 0.39 to 0.82 with BC alone or BC plus dust,
and from 0.70 to 0.85 with dust alone; the corresponding radiative forcing was 3.93–43.44 Wm$^{-2}$ for BC alone, 4.01–
43.45 Wm$^{-2}$ for BC and dust, and 0.16–2.08 Wm$^{-2}$ with dust alone. The radiative forcing calculated from the daily
mean albedo reduction ranged from 0.1% for dust only at the low pollutant site to 14.9% for BC at the high pollutant
site. The potential source regions of the pollutants deposited on glaciers and snow were identified using spatial
variance in wind vector maps, emission inventories coupled with back air trajectories, and region tagged chemical
transport modelling. The wind vector maps identified Central Asian and South Asian countries (such as Azerbaijan,
Turkmenistan, Pakistan, Afghanistan, Syria, Iraq, Turkey) as more important. The trajectory analysis coupled with
emission inventories showed that air parcels reached northern Pakistan along three pathways, one from north Asia
(Russia) via Central Asia (Kazakhstan), one from western Asia (Cyprus and Syria) via Central and southern Asia
(Georgia), and one via India. Combination of the back-trajectory results and surface-wind direction analysis
indicated that aerosols were significantly influenced by the long-range transport of pollutants from Central and
South Asia. The region-tagged chemical transport model indicated that Pakistan and South Asia were the main
contributors of pollutants. Analysis based on the WRF-STEM model identified a significant contribution from
Pakistan (up to 77%) and South Asia (up to 71%) at selected sites. Overall, the results indicate that Central, South,
and West Asia were the major sources of the pollutants detected at the sampling sites, with only a small contribution
from East Asia.

The overall uncertainty of the BC, and OC concentrations was estimated taking into account the analytical

precision of concentration measurements and the mass contribution from field blanks. The uncertainty in the BC and
OC mass concentrations was calculated from the standard deviation of the field blanks, the experimentally
determined analytical uncertainty, and the projected uncertainty associated with filter extraction. The major source
of uncertainty was the effect of dust on the OC/BC measurements.

The albedo reduction from OC was not quantified. The contribution of OC to total visible absorption in the top

snow layer is relatively small compared to that of BC and dust but has been shown to be significant (~19% of the
total solar visible absorption) in several regions including northeastern East Asia, and western Canada (Yasunari et
al., 2015). Snow grain size (snow aging) and snow texture were probably the main sources of uncertainty in the
albedo reduction/radiative forcing calculations. The measured grain size was generally different from the effective
optical grain size used in the SNICAR modeling, and although snow grain shape was measured, the results were not
used in the online SNICAR albedo simulation model, which assumes a spherical shape for snow grains. This could
slightly affect the results, because the albedo of non-spherical grain is higher than the albedo of spherical grains
(Dang et al., 2016).

The possible uncertainties on the modeling side relate to the use of CO as a tracer for light absorbing particles to

identify the source region. Uncertainties are also attributed to errors in the emission inventories, simulated
meteorology, and removal processes built into the model. The physics and chemistry of removal of BC and -CO
differ, especially in the wet season. However, we analyzed the model during pre-monsoon and relatively dry periods
when there should be a relatively good correlation in the transport of CO and BC. The global emission inventories
used are unable to capture emissions at a local scale, and the contribution of local sources may also be
underestimated by coarse-resolution models. High resolution models and emission inventories at a local scale are
required to capture local emissions.

Better constrained measurements will be required to obtain more robust results. High resolution satellite imagery,

high resolution models, and continuous monitoring will help to reduce the present uncertainty.
**Acknowledgments**

This study was supported by the National Natural Science Foundation of China (41630754, 41671067,

41721091), the Chinese Academy of Sciences (QYZDJ-SSW-DQC039), the State Key Laboratory of Cryosphere
Science (SKLCS-ZZ-2017), program funding to ICIMOD from the Governments of Sweden and Norway, and
ICIMOD core funds contributed by the Governments of Afghanistan, Australia, Austria, Bangladesh, Bhutan, China,

India, Myanmar, Nepal, Norway, Pakistan, Switzerland, and the United Kingdom. Acknowledgement is also due to Dr A Beatrice Murray for English editing of the manuscript. The authors would like to thank both the anonymous reviewers, whose reviews were extremely helpful in enhancing the quality of the manuscript. We would also like to convey our gratitude to the editor for smooth handling of the manuscript.

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

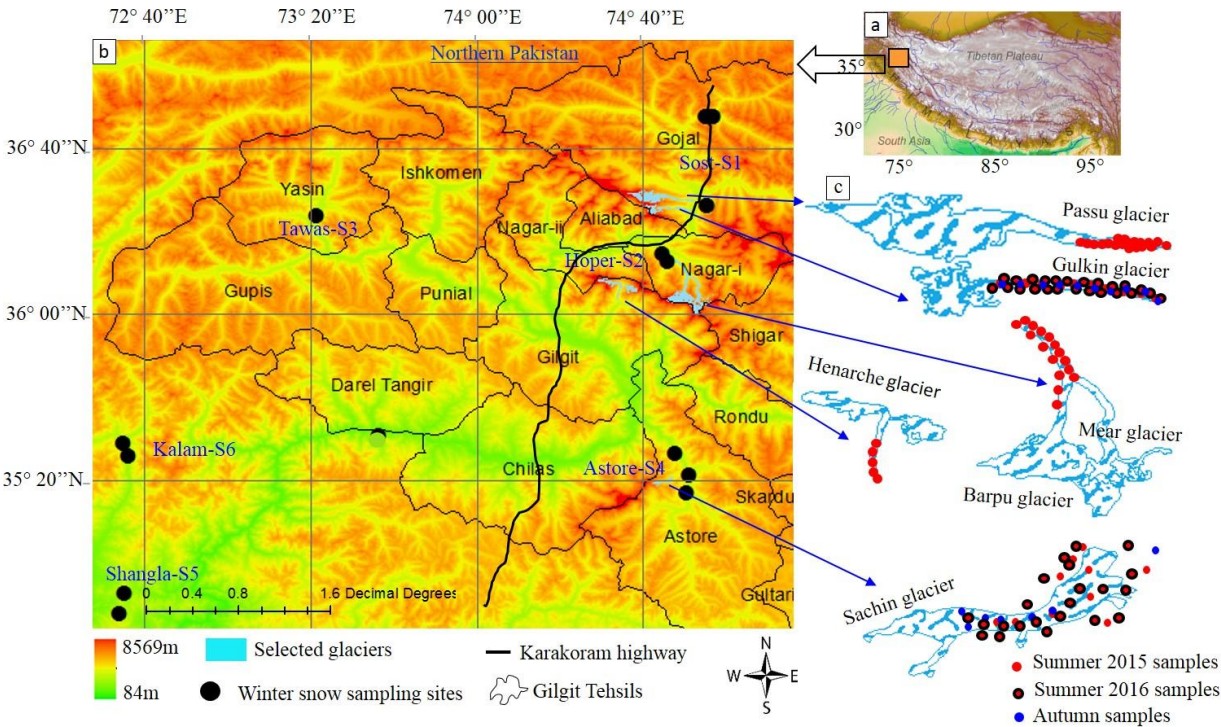


**Figure 1. The study area and sampling sites: (a) Himalayan mountain range and Tibetan Plateau, (b) winter sampling sites (solid black circles), (c) glaciers selected for**
**summer and autumn sampling**


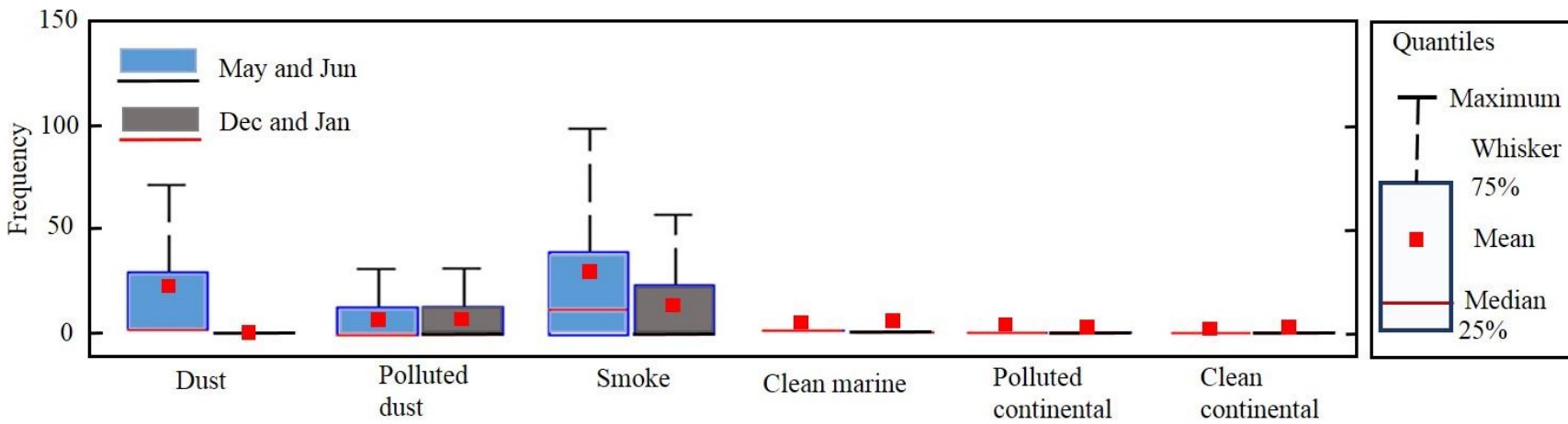


**Figure 2. Frequency distribution of aerosol subtypes in the atmosphere over the study region calculated from CALIPSO data, average for the study months in 2006 to**
**2014**

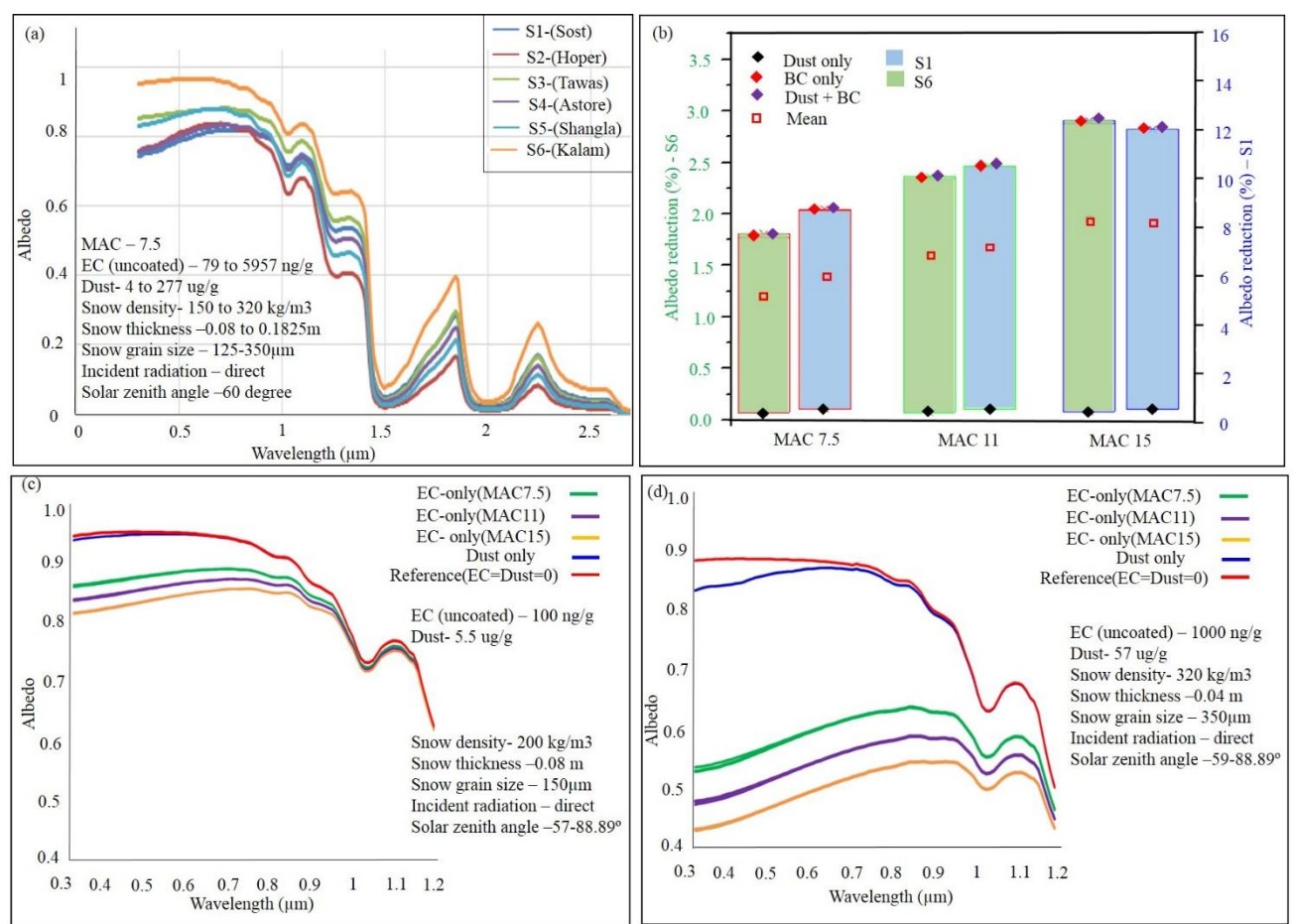


**Figure 3. Spectral variation in albedo for winter sampling sites and selected mass absorption cross-section (MAC) values, (a) average albedo of samples at each of the sites**
**(b) daily mean albedo reduction of fresh snow (site S6) and aged snow (site S1) snow, (note different scales of y axis) (c) albedo of fresh snow site S6, (d) albedo of aged**
**snow site S1**


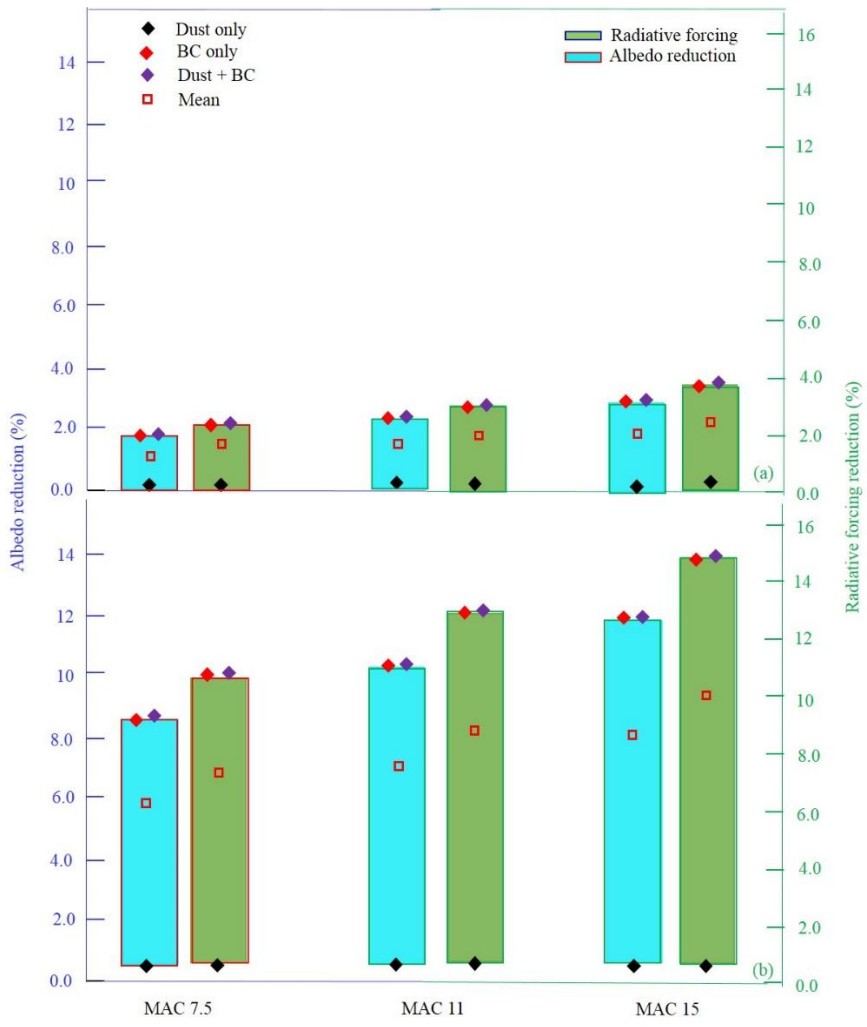


**Figure 4. Daily mean radiative forcing reduction and albedo reduction (%) caused by black carbon and dust for different mass absorption cross-section values (MAC) in (a) fresh (low black carbon) and (b) aged (high black carbon) snow samples (note different scales of y axis)**


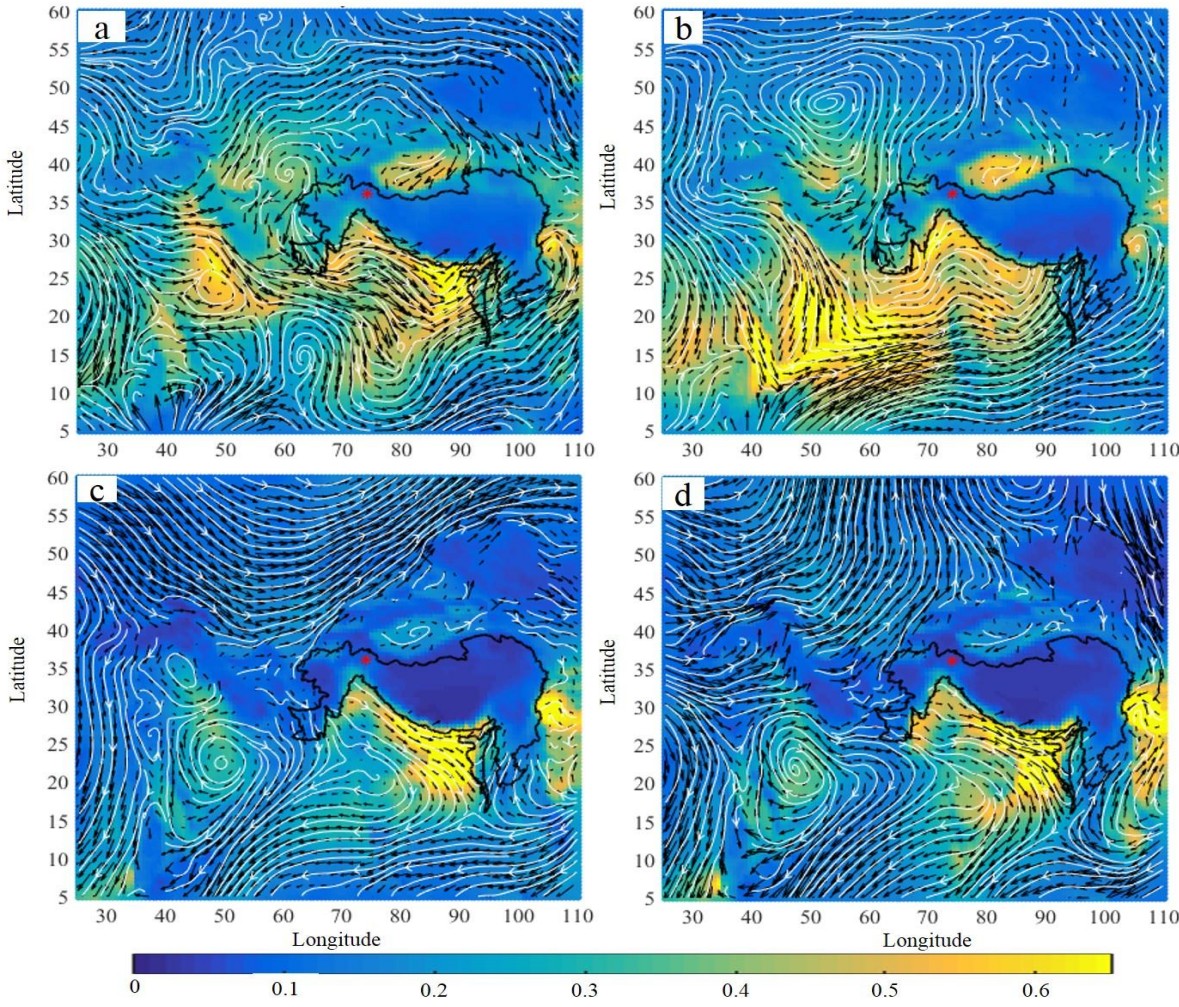


**Figure 5. Monthly average horizontal wind patterns at 850 hPa during a) May, b) June, c) December, and d) January, corresponding to approximately 2,500 masl, from**
**GES DISC. Red star indicates the position of the study area, and white lines indicate streamlines. The background colors show monthly mean aerosol optical depth.**


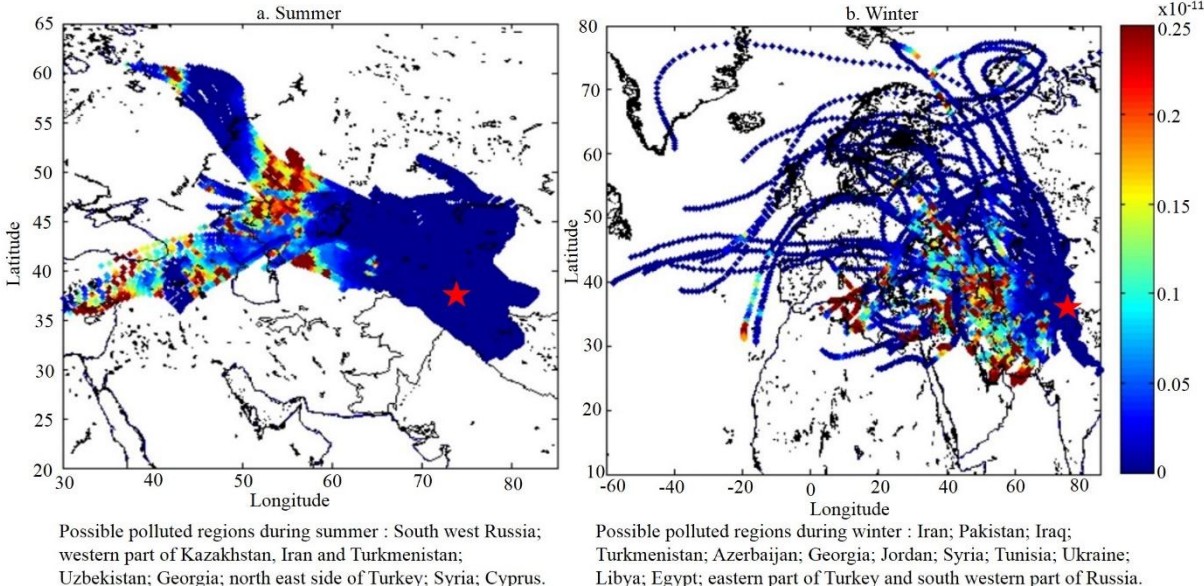

Possible polluted regions during summer : South west Russia; western part of Kazakhstan, Iran and Turkmenistan; Uzbekistan; Georgia; north east side of Turkey; Syria; Cyprus.

Possible polluted regions during winter : Iran; Pakistan; Iraq; Turkmenistan; Azerbaijan; Georgia; Jordan; Syria; Tunisia; Ukraine; Libya; Egypt; eastern part of Turkey and south western part of Russia.


**Figure 6. Source contribution regions of pollutants identified using an emissions inventory (Representative Concentration Pathways) coupled with back trajectories: a) 77**
**simulated days, b) 63 simulated days. Red star indicates the position of the study area.**

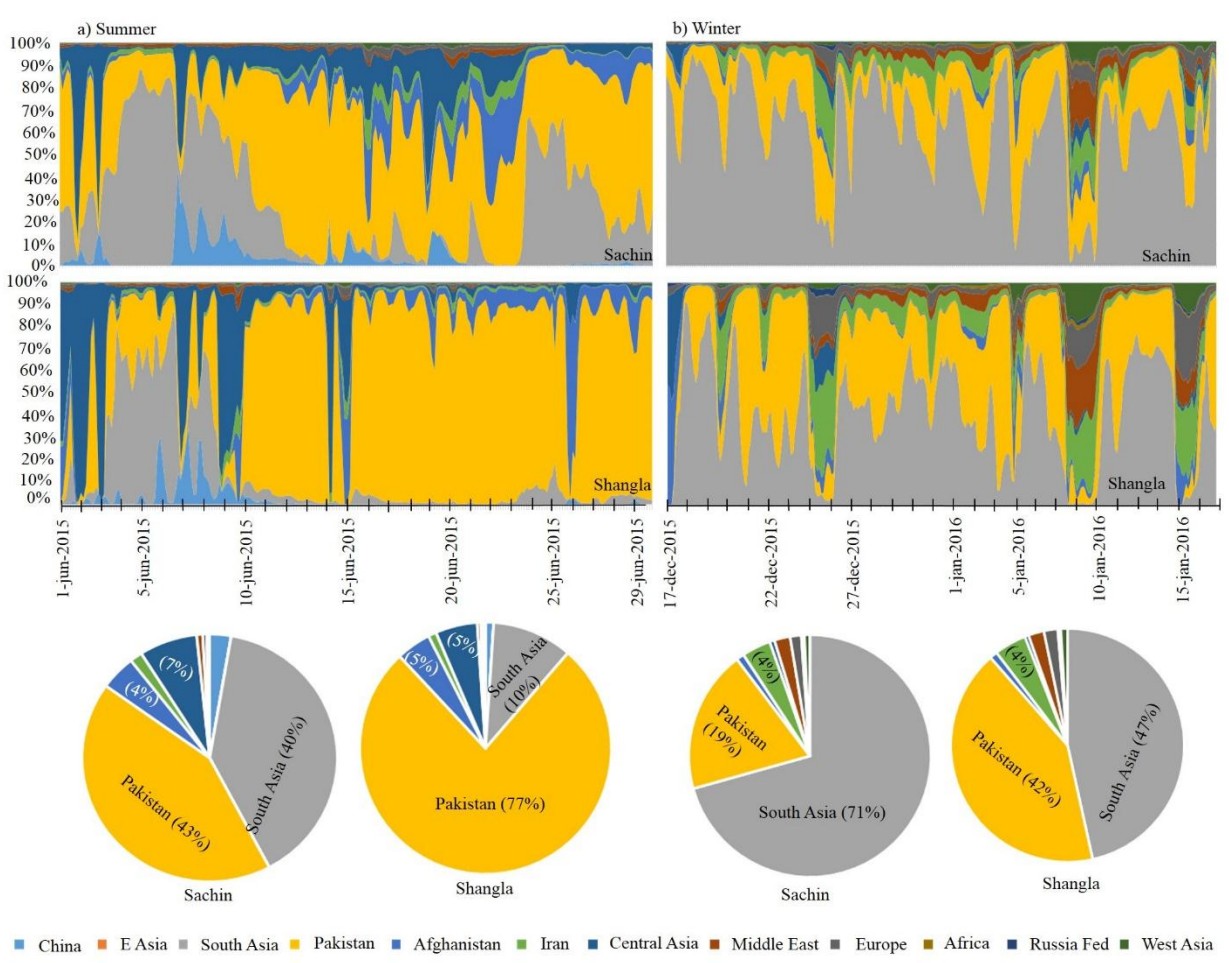


**Figure 7. Source contribution regions of carbon monoxide for selected sites identified by WRF-STEM during (a) summer and (b) winter**

**Table 1. Concentration of black carbon, organic carbon, and dust in summer, autumn, and winter samples in 2015 and 2016**

| Glacier/ Site | No. min–max | Elevation (masl) | BC min–max (avg) (ng g⁻¹) | OC min–max (avg) (ng g⁻¹) | Dust min–max (avg) (µgg⁻¹) | Type[a] / snow age in days | OC/BC[b] | Year |
|---|---|---|---|---|---|---|---|---|
| **Summer (May 2015/ May 2016)** | | | | | | | | |
| **Barpu** | 6 | 2901–3405 | 877–5994 (2938) | 244–1228 (691) | 292–5250 (1998) | DCI | 0.07–1.38 | 2015 |
| **Gulkin** | 31 | 2741–3319 | 82–5676 (1327) | 238–8514 (1594) | 31–2039 (648) | DCIS | 0.169–3.76 | 2015/16 |
| **Henarche** | 4 | 2569–2989 | 778–10502 (4820) | 275–4176 (1628) | 225–2723 (993) | Ice | 0.04–1.63 | 2015 |
| **Mear** | 8 | 2961–3539 | 222–3656 (1593) | 703–6588 (2992) | 33–656 (211) | DCI | 0.72–4.88 | 2015 |
| **Passu** | 14 | 2663–3158 | 87–734 (346) | 132–1810 (741) | 28–524 (196) | DCI | 1.85–4.80 | 2015 |
| **Sachin** | 35 | 3414–3895 | 257–4127 (1769) | 128–7592 (3348) | 5.6–2495 (314) | DCIS | 0.08–0.53 | 2015/16 |
| **Total** | 98 | | | | | | | |
| **Autumn (October 2016)** | | | | | | | | |
| **Gulkin** | 7 | 2741–3319 | 125–1028 (451) | 266–3574 (1276) | 60–767 (253) | DCIS | 1.29–3.59 | 2016 |
| **Sachin** | 6 | 3414–3895 | 4342–6481 (5314) | 543–3478 (1571) | 124–1348 (546) | DCIS | 0.11–0.53 | 2016 |
| **Total** | 13 | | | | | | | |
| **Winter (Dec 2015/ Jan 2016)** | | | | | | | | |
| **S1-Sost** | 6 | 2873–3092 | 482–5957 (2506) | 378–2934 (1039) | 29–311 (131) | 2–17 d | 0.25–0.78 | 2015 |
| **S2-Hopar** | 2 | 2602–2794 | 229–1064 (646) | 330–1976 (1153) | 23–129 (76) | 1–15 d | 1.4–1.8 | 2016 |
| **S3-Tawas** | 1 | 2437 | 650 | 1320 | 16 | 8–17 d | 2.03 | 2016 |
| **S4-Astore** | 3 | 2132–2396 | 450–2640 (1305) | 914–3645 (2161) | 55–171 (97) | 4–7 d | 1.38–2.33 | 2016 |
| **S5-Shangla** | 2 | 2324–2373 | 367–1110 (739) | 1302–2856 (2079) | 13–49 (31) | 8–9 d | 2.5–3.5 | 2016 |
| **S6-Kalam** | 4 | 1933–2101 | 79–123 (107) | 214–558 (347) | 4–6 (5) | 1 d | 2.3–5 | 2016 |
| **Total** | 18 | | | | | | | |

[a] type = snow or ice type; DCI = debris-covered ice; DCIS = debris-covered ice and aged snow
[b] range of OC/BC in individual samples

**Table 2. Snow albedo reduction (%)by black carbon, dust, and black carbon plus dust at the site with the lowest average pollutant concentration (S6) and the site with the**
**highest average pollutant concentration (S1), under different mass absorption cross-section (MAC) values**

| Pollutant | MAC value (m²/g) | Low concentration site (S6) | | | High concentration site (S1) | | |
|---|---|---|---|---|---|---|---|
| | | Daytime min | Daytime max | Daily mean | Daytime min | Daytime max | Daily mean |
| **Black carbon** | 7.5 | 2.8 | 5.1 | 1.8 | 15.6 | 23.9 | 9.0 |
| | 11 | 3.7 | 6.9 | 2.3 | 19.2 | 28.6 | 10.5 |
| | 15 | 4.6 | 8.3 | 2.9 | 22.3 | 32.5 | 12.0 |
| **Dust** | 7.5 | 0.1 | 0.2 | 0.07 | 0.9 | 1.6 | 0.05 |
| | 11 | 0.1 | 0.2 | 0.07 | 0.9 | 1.6 | 0.05 |
| | 15 | 0.1 | 0.2 | 0.07 | 0.9 | 1.6 | 0.05 |
| **Black carbon and dust** | 7.5 | 2.9 | 5.2 | 1.8 | 15.7 | 24.0 | 8.8 |
| | 11 | 3.8 | 6.8 | 2.4 | 19.2 | 28.6 | 10.5 |
| | 15 | 4.6 | 8.3 | 2.9 | 22.3 | 32.5 | 12.0 |
