# Peer review of "Concentrations and source regions of light absorbing particles"

_Atmospheric Chemistry and Physics, 2017_

## Referee Comment (RC1) · Anonymous Referee #2 · 26 Sep 2017

This study is valuable because it describes measurements of black carbon in snow from the Karakoram/Himalayan region of Pakistan. To my knowledge these are the first such measurements to be reported from this region. Moreover, the reported concentrations of BC in snow are extremely large, indicative of pollution being a major source of snow/ice albedo reduction in this area. The study is also comprehensive in the sense that it applies CALIPSO observations of aerosol type, back-trajectory analysis, and regional chemistry/climate modeling to ascertain dominant sources of pollution to the snow and glaciers in the study area. Despite the value of having new measurements from the Karakoram, a region with a paucity of environmental data, the study has some weaknesses that are described below. Ultimately, I believe these issues lead to conclusions which are somewhat vague. I suppose the main take-home message, however, is that there is a lot of BC in low elevation glaciers and snow of northern Pakistan, and perhaps this is a sufficient conclusion in and of itself for publication. Below, however, are the major issues I see with the current draft of the paper.

(1) The CALIPSO aerosol source identification analysis indicates that "smoke" is the most frequently-occurring type of aerosol over this region during both summer and winter. As the authors acknowledge, however, biomass burning sources were not included in the WRF-STEM modeling, and thus the dominant source regions identified through the WRF modeling may not be representative at all for the BC that was measured. Moreover, were biomass burning sources included in the RCP emission inventory that was utilized with the back-trajectory analysis? (Please include more information about the RCP emissions that were used.) A third question related to the source attribution analysis is: Potentially how important are local (e.g., within ~10km) sources occurring within the same "grid cell" of the WRF and HYSPLIT models? Contributions of such local (sub-grid scale) sources may be severely underestimated by coarse-resolution models. Some of the discussion suggests that local sources may have been very important, but these sources did not really enter into the assessment (via HYSPLIT and WRF) of source attribution.

(2) The values of BC in snow that were found are extremely large, but it is also acknowledged in the paper that the measurements were taken close to sources of pollution, namely roadways and villages. I am left wondering how representative the reported snow pollution values are of the broader Karakoram cryospheric region. The answer to this may not be known, but some discussion, even if speculative, about this issue would be appreciated. Do these measurements suggest that the glaciers of the Karakoram, in general, are being substantially darkened by BC, or do they simply mean that the ablation zones of a few glaciers near to obvious BC sources are quite polluted?

(3) The authors report that "there was no clear correlation between BC and OC concentrations" (line 269), which I found a bit worrisome given that the two species usually
originate from common sources and have common transport pathways. The authors do provide some potential reasons for why we could find more BC than OC in the snow (e.g. ,greater melt scavenging of OC), which was also a bit surprising, but I would appreciate seeing some more discussion on why concentrations of BC and OC would be uncorrelated.

(4) Related to the point above, how precisely was OC differentiated from BC in the thermal optical technique? What temperature threshold or thermal evolution profile was applied to separate the two species? Could this have had anything to do with the high BC/OC ratios that were found in the snow samples?

(5) More generally, please describe and if possible quantify, sources of uncertainty in the measurements of BC, OC, and dust in snow.

(6) My sense is that snow grain size and snow texture are larger sources of uncertainty in the albedo reduction / RF calculations than indicated. Although snow grain size was measured with a hand lens (with reported accuracy of 20um), this determination of grain size is usually different from the effective (surface area-weighted) / optical grain size used in the SNICAR modeling. The true uncertainty in effective/optical grain size is likely much larger than 20um, and I think the paper should include greater acknowledgment of this issue. The discussion of albedo variability associated with snow grain size (or snow aging) should also more clearly indicate the ranges in snow grain size that were assumed for the albedo modeling.

Furthermore, references to "snow age" are sometimes used when "snow grain size" would be more appropriate, since snow grain size does not always increase monotonically with snow age, and it is really the snow grain size that matters for optical/radiative considerations. Examples of this is are on line 364: "The estimated reduction in snow albedo by dust and BC compounded by the age of snow..." and line 386: "... exact snow age ...".

(7) Snow albedo and perturbations to albedo are modeled and used heavily in this

**ACPD**
study to derive radiative forcing estimates, but no observations of snow or ice albedo are reported. Are there any observations of snow and ice albedo from this region that could be utilized to help verify or support the modeling? I worry in particular that debris could strongly reduce albedo of the glaciers but is neglected in the model, potentially leading to bias in the modeled albedo perturbations.

Minor issues:

line 211: "... were put in the above equation and got a normalized extinction..." - grammar issue.

line 251: "... with the generally lower deposition on the Gulkin glacier more affected by other factors" - Which factors?

line 257: "is considering as" -> "considered as"

line 274: "... low OC/BC ratios can result from a reduction in OC, greater contributions from BC enrichment..." - It is unclear to me which processes "reduction in OC" and "BC enrichment" refer to. Could the authors please elaborate on these processes?

line 329: "albedo of samples from the two sites simulated at a wavelength of 0.975um ... " - Why are 0.975um albedo values reported here? Light-absorbing impurities exert the strongest influence on blue or mid-visible albedo (e.g.,  $\sim$ 0.450um). The 0.975um albedo is affected less strongly by impurities, and somewhat heavily by snow grain size, so it seems an odd choice of wavelength to use for reporting albedos.

line 336: "The results suggest that BC was the dominant forcing factor, rather than dust, as a result of the rapid snowmelt." - The identification of "rapid snowmelt" as the cause for greater BC forcing than dust forcing is confusing here. Perhaps the sentence just needs re-working. Otherwise, what role does snowmelt play in the determination of instantaneous radiative forcing?

line 343: "... reduction in daily mean albedo of 1.8 to 2.9% ... " Are these relative or absolute reductions in albedo? If the latter, please use absolute (non-percentage)

**ACPD**
units. This also applies to other references to percent albedo reduction in the paper.

lines 400-401: Which environments do these RF estimates apply to?

lines 406-410: It should be acknowledged again that dust forcing varies strongly with dust optical properties and particle size distribution. The estimates derived here appear to have utilized a generic representation of dust in the model that may or may no be appropriate for the dust that was actually measured.

lines 461: "BC from East Asia can potentially be lifted up high and transported to the northeast during the summer monsoon season. Nonetheless..." - But transport of East Asia emissions to the northeast does not seem relevant for deposition in Pakistan. Please clarify the relevance of this statement.

line 463: "... low latitude source regions such as South Africa..." - I suggest using "tropical Africa" or something similar here instead of "South Africa" (which happens to also be the name of a country).

line 464: "weak emissions" - Actually, biomass burning emissions from tropical regions of Africa constitute a substantial share of global BC emissions, so "weak" may not be the best word here.

line 479: "considerable" -> "considerably"

line 482: "The concentration of hydrophobic BC, hydrophilic BC, ... " - The description earlier in Methods indicated only that CO tracers were used. Was BC also simulated with this model? If so, were BC tags applied? Please include more description of the BC simulation in Methods. This seems much more relevant for source attribution, since the physics and chemistry of removal for BC and CO are quite different from each other.

line 520: "... and increased grain size and density." - It is not clear to me how snow grain size and snow density should affect the \*concentration\* of BC, as indicated in this sentence. Please clarify.

ACPD
Figure 2: Most of the figure is white space. I suggest shrinking the y-axis range to show the plot values more clearly.

Figure 6: Please specify which emission inventory was used and how many days of back-trajectory were simulated.

Table 2: Are these relative or absolute snow albedo reductions?

---

## Referee Comment (RC2) · Anonymous Referee #1 · 14 Nov 2017

This manuscript reported the data of light absorbing particles (LAP, such BC, OC and dust) measured in snow/ice in northern Pakistan and estimated the induced snow albedo reduction and corresponded radiative forcing during 2015-2016. Authors found the concentration of BC, OC and dust in aged snow is higher than in fresh snow and ice and the concentration over northern Pakistan is higher than over the Himalayas and Tibetan Plateau. Estimated LAP-induced daily mean snow albedo reduction is approximately 0.07-12.0% and corresponded radiative forcing is approximately 0.16-43.5 W m-2, depending on snow type, solar zenith angle, and locations. Also different methods are used in this study to identify the source regions of pollutants measured in this region.

[Figure]

Overall the results are interesting and measurement data are valuable for the community. The figures and tables in the manuscript are relevant, but not very good in quality, and need to be improved. In general the paper is well written but in many places the English could and should be improved. There are some major weaknesses in the manuscript, especially in the source region identification part (3.5) and aerosol type frequency distribution part (3.2). After the below comments are appropriately addressed, I would suggest to accept the manuscript for publication in the Atmospheric Chemistry and Physics.

Technical comments:

Introduction, I would suggest use Light-absorbing particles (LAP) instead of Light-absorbing impurities, see Qian et al., 2015, which by the way is a review article for both measurement methods and modeling activities. This article is so relevant so probably should be cited in Introduction part.

Lines 62-65, Besides warming efficiency, another important characteristic of LAPs is its higher snowmelt efficiency, see Qian et al., 2011.

Lines 110, precipitation 0.412 +/- 2 mm for per day or per year?

Lines 123-128, again sample method is summarized in Qian et al., 2015.

Section 2.3.1, without 2.3.2, really needs 2.3.1? More details are needed regarding how the L-2 data are processed.

Lines 155-159, This paragraph should be removed or moved to Introduction section.

Section 2.5 is very poorly organized and kind of just present whatever tools you have or used before, without a clear goal or coherence in science structure. Must be rewritten.

2.5.1 Wind maps, why not use 2005-2006 wind maps instead of 50-year average? 700mb is very high level for low-elevation region and aerosol concentration is very low at that high level. I would suggest use the terrain-oriented level like sigma level near the

surface. I would also strongly suggest (a must do) use MERRA-2 reanalysis data, in which not only the data quality is better than NCEP/NCAR but also it includes aerosol data that can be used to compare with the measurement and is more appropriate for looking at the long-distance transport.

2.5.3, WRF-STEM can only tag CO, because of many differences between CO and LAP in such as emission sources, chemistry and removal, how to quantify their differences in long-range transport and source identifications? How to infer the transport and source for LAPs based on CO and what's the uncertainty? Please see Zhang et al. 2015 and Wang et al., 2015 for source detection methods used over Tibetan Plateau region.

Line 257, 24 hours. Considering–>considered.

Line 318, Jun–>June.

Section 3.2, very weak! How to connect the conclusion from this section with other parts?

Section 4 Summary and conclusion, one more section should be added for discussion in uncertainty and possible future direction for both modeling and measurement campaign. For example, how snow aging (snow grain size) and melting water scavenging efficiency (see Qian et al, 2014) affect the conclusions?

Figure 2, this is a poor figure and should be re-designed. For example, reduce the y-axis range from 300 to 150. Btw why the numbers for y-axis are 50, 100, 150, 200, 150 (should be 250?)?

Figure 3, give full name for MAC in figure caption. Also consider use identical range for y-axis e.g. 0.4-1.0 and for x-axis 0.3-1.2 for Panel c and d.

Figure 4, suggest use identical y-axis range so can highlight the bigger effect over aged snow. The unit for radiative forcing is %? More discussion should be provided regarding how snow aging affect the albedo reduction and radiative forcing (e.g. Qian

et al., 2014)?

Figure 5, what blue contours represent? Again 700 mb is too high and MERRA-2 is a much better dataset.

Figure 6, not clear what color shades represent?

Figure 7, I am not sure how the quantitative number of contributions are meaningful because the numbers for LAP could be very different with that for CO. Anyway again, Section 2.5 is very poorly organized and kind of just present whatever tools you have or used before, without a clear goal or coherence in science structure. Need to be rewritten with a clear conclusion.

Table 2, give full name for MAC (in other tables/ figures as well).

Figure S7, give full names for BC1 and BC2.

References: Qian Y, TJ Yasunari, SJ Doherty, MG Flanner, WK Lau, J Ming, H Wang, M Wang, SG Warren, and R Zhang. 2015. "Light-absorbing Particles in Snow and Ice: Measurement and Modeling of Climatic and Hydrological Impact." Advances in Atmospheric Sciences 32(1):64-91. doi:10.1007/s00376-014-0010-0. Zhang R, H Wang, Y Qian, PJ Rasch, RC Easter, Jr, PL Ma, B Singh, J Huang, and Q Fu. 2015. "Quantifying sources, transport, deposition, and radiative forcing of black carbon over the Himalayas and Tibetan Plateau." Atmospheric Chemistry and Physics 15(11):6205-6223. doi:10.5194/acp-15-6205-2015. Qian Y, H Wang, R Zhang, MG Flanner, and PJ Rasch. 2014. "A Sensitivity Study on Modeling Black Carbon in Snow and its Radiative Forcing over the Arctic and Northern China." Environmental Research Letters 9(6):Article No. 064001. doi:10.1088/1748-9326/9/6/064001. Wang M, B Xu, J Cao, X Tie, H Wang, R Zhang, Y Qian, PJ Rasch, S Zhao, G Wu, H Zhao, DR Joswiak, J Li, and Y Xie. 2015. "Carbonaceous Aerosols Recorded in a Southeastern Tibetan Glacier: Analysis of Temporal Variations and Model Estimates of Sources and Radiative Forcing." Atmospheric Chemistry and Physics 15:1191-1204. doi:10.5194/acp-15-

1191-2015. Qian Y, MG Flanner, LYR Leung, and W Wang. 2011. "Sensitivity studies on the impacts of Tibetan Plateau snowpack pollution on the Asian hydrological cycle and monsoon climate." Atmospheric Chemistry and Physics 11(5):1929-1948. doi:10.

---

## Author Comment (AC1) · 10 Jan 2018

**Major issues:**

This study is valuable because it describes measurements of black carbon in snow from the Karakoram/Himalayan region of Pakistan. To my knowledge these are the first such measurements to be reported from this region. Moreover, the reported concentrations of BC in snow are extremely large, indicative of pollution being a major source of snow/ice albedo reduction in this area. The study is also comprehensive in the sense that it applies CALIPSO observations of aerosol type, back-trajectory analysis, and regional chemistry/climate modeling to ascertain dominant sources of pollution to the snow and glaciers in the study area. Despite the value of having new measurements from the Karakoram, a region with a paucity of environmental data, the study has some weaknesses that are described below. Ultimately, I believe these issues lead to conclusions which are somewhat vague. I suppose the main take-home message, however, is that there is a lot of BC in low elevation glaciers and snow of northern Pakistan, and perhaps this is a sufficient conclusion in and of itself for publication. Below, however, are the major issues I see with the current draft of the paper.

**Response:**

We thank the reviewer for their comments that significantly contributed to improving the original manuscript. Please see below our comment-by comment-responses to each of the reviewer's comments and suggestions.

Reviewer Comments in black

Responses in blue

Modified text in the revised manuscript is in green.

1. **(1a) The CALIPSO aerosol source identification analysis indicates that "smoke" is the most frequently-occurring type of aerosol over this region during both summer and winter. As the authors acknowledge, however, biomass burning sources were not included in the WRF-STEM modeling, and thus the dominant source regions identified through the WRF modeling may not be representative at all for the BC that was measured.**

   **(1b)Moreover, were biomass burning sources included in the RCP emission inventory that was utilized with the back-trajectory analysis? (Please include more information about the RCP emissions that were used.)**

**(1c)A third question related to the source attribution analysis is: Potentially how important are local (e.g., within _10km) sources occurring within the same "grid cell" of the WRF and HYSPLIT models? Contributions of such local (sub-grid scale) sources may be severely underestimated by coarse-resolution models. Some of the discussion suggests that local sources may have been very important, but these sources did not really enter into the assessment (via HYSPLIT and WRF) of source attribution.**

Response:

(1a). Major part from biomass burning sources (biofuel) were included in the WRF-STEM modeling, and we think the dominant source regions identified through the WRF modeling should represent majority of the BC (pollutants) regions that was measured. Apologies for not mentioning these important information in our initially submitted manuscript.

The sentence related to biomass burning has been modified in the revised manuscript (lines 260-266), given below for your reference.

"The Hemispheric Transport Air Pollution (HTAP version 2) emission inventory was used in our WRF-STEM modeling. The HTAP version 2 dataset consists of multiple pollutants including black carbon and organic carbon. All type of biomass burning (such as energy, industry, transport, residential etc...) are included in HTAP emission inventory (except large scale open agricultural and open forest fire burning). The simulations applied in our study used the anthropogenic emissions from HTAPv2 inventory (available from http://edgar.jrc.ec.europa.eu/htap_v2/). So the results indicate the amount of pollutants reaching the study area from day-to-day planned and recurring activities in domestic, transport, industrial, and other sectors."

(1b) Yes biomass burning sources were included in the RCP emission inventory that was utilized with the back-trajectory analysis. Related information about the used RCP emissions has been added in the revised manuscript (line 230-239), quoted below for your reference.

"The data file used as a RCP emission inventory was "RCPs_anthro_BC_2005-2100_95371.nc". This comprises emissions pathways starting from identical base year (2000) for multiple pollutants including black carbon and organic carbon. According to the description of the file, biomass burning sources were included in the RCP emission inventory that were utilized with the back-trajectory analysis. RCP had the same emissions sectors as for HTAP emission inventory used in the molding part. The emission sectors includes fuel combustion, industries, agriculture and livestock. The difference in HTAP and RCP emission inventories is the resolution. HTAP had relatively high resolution (0.1 x 0.1 degree) as compared to RCP (0.5 x 0.5 degree). Some discussion related to the inventory and the sectorial detail (12 sectors), which was used for the base year calibration of the RCPs is given in Lamarque et al., 2010."

(1c) Local sources and local emissions may have importance, but based on available options it was hard to capture. We are expecting minor impact of local emissions due to below reasons.

- There were limited transport on Karakorum highway and sparse residential houses in surrounding region (within_10km), near the glaciers.
- The glaciers in the surrounding region had relatively high altitude and away from main urban emission sources and urban areas.

There may be the slight effect of local transport, house cooking but we were unable to capture that local scale emissions. The chemical transport model (WRF-STEM) and RCPs were based on emission inventories and does not capture/does not collect the local emissions. In order to reduce uncertainty in source region high resolution BC tracer will be used in our next publication in near future.

Source contribution regions of pollutants identified using an emissions inventory (Representative Concentration Pathways) are shown in Figure 1 below. Lower part of the figure indicating local/regional source regions within 221Km x 276 Km region. As the resolution of RCP emission data is 0.5 x 0.5 degree so there is no change within 55 x 55 $km^2$ area. Using global emission inventories we are unable to capture emissions at local scale (within 10 km region). High resolution models and emission inventories at local scale are required to capture local emissions. Below text has been added in lines 675-678.

"While using global emission inventories we were unable to capture emissions at local scale. Contributions of local sources may be underestimated by coarse-resolution models. Therefore high resolution models and emission inventories at local scale are required to capture local emissions."

[Figure]

Figure1: Source contribution regions of pollutants identified using an emissions inventory (Representative Concentration Pathways). Red stars indicating sampling locations.

2.  (a) **The values of BC in snow that were found are extremely large, but it is also acknowledged in the paper that the measurements were taken close to sources of pollution, namely roadways and villages.**
    (b) **I am left wondering how representative the reported snow pollution values are of the broader Karakoram cryospheric region. The answer to this may not be known, but some discussion, even if speculative, about this issue would be appreciated. Do these measurements suggest that the glaciers of the Karakoram, in general, are being substantially darkened by BC, or do they simply mean that the ablation zones of a few glaciers near to obvious BC sources are quite polluted?**

Response: (a) High concentration of BC in snow and ice:

The value of BC in snow and ice that was found relatively high and we justified it in our manuscript as given below for your reference.

- ➢ Sampling locations were relatively at lower elevation as compared to other studies in the past (lines 446-447). Li et al. (2017) showed a strong negative relationship between the elevation of glacier sampling locations and the concentration of light absorbing particles (lines 365-366).
- ➢ Majority of samples were from the ablation zone of the glaciers. Strong melting of surface snow and ice in the glacier ablation zone could also lead BC enrichment which causes high BC concentrations as Li et al., 2017 observed in the Southern Tibetan Plateau glacier (lines 447-448).
- ➢ In most cases snow and ice samples were collected quite a long time after snow fall, and the concentration of pollutants would also have increased in the surface snow and ice due to dry deposition (lines 352-353).
- ➢ In the past almost similar high concentration were reported by multiple authors in the region such as Xu et al. 2012 in the Tien Shan Mountains, Li et al. 2016 in the northeast of the Tibetan plateau, Wang et al. 2016 in northern China, Zhang et al., 2016 in southeastern Tibetan plateau and Zhang et al. 2017 in western Tien Shan, Central Asia (lines 341-343).

(b) Glaciers of the Karakoram, in general, are being substantially darkened by BC?

According to our understanding all the glaciers of the whole Karakoram region, may not besubstantially darkened by BC, as in case of our selected glaciers. On the basis of limited samples from selected glaciers, it is hard to conclude a general statement to represent the whole Karakorum region. Further research based on in-situ observations, satellite based observation and high resolution modeling and emission inventories are required. We are expecting that ablation zones of the debris covered glaciers which are relatively at low elevation and near to pollution sources may be quite polluted, especially during melting seasons (we have updated this information in lines 372-374, given below for your reference).

"According to our understanding all the glaciers of the whole Karakoram region, may not besubstantially darkened by BC. Ablation zones of the debris covered glaciers which are relatively at low elevation and near to pollution source may be quite polluted."

3. **The authors report that "there was no clear correlation between BC and OC concentrations" (line 269), which I found a bit worrisome given that the two species usually originate from common sources and have common transport pathways. The authors do provide some potential reasons for why we could find more BC than OC in the snow (e.g. ,greater melt scavenging of OC), which was also a bit surprising, but I would appreciate seeing some more discussion on why concentrations of BC and OC would be uncorrelated.**

Response:

Yes the concentration of BC and OC was uncorrelated. In most cases the concentration of OC was greater than the concentration of BC. In few cases the concentration of BC was greater than the concentration of OC concentration. We add an additional text in revised manuscript lines 303-315, given below for your reference.

"In most cases the concentration of OC was greater than the concentration of BC. In few cases the concentration of BC was greater than the concentration of OC, which might indicates the contribution of coal combustion and/or biomass burning to the emissions. The reported OC concentration was water-insoluble OC. Including the water soluble OC could dominate the temporal variation of the OC/BC ratio. One important factor was post-deposition process, melt water can bring dissolved organic carbon away but not for BC. Low OC/BC ratio may also be possible due to the fact that OC and BC had redistributed primarily under the control of strong melt water rather than sublimation and/or dry/wet deposition. The spatio-temporal variability of OC/BC ratio may also indicate the contribution of various sources, seasonal variation and frequent change in wind directions. The OC vs BC correlation in snow and ice samples depend on OC vs BC ratio/concentrations in the atmosphere, post deposition process and then scavenging, enrichment and melt rate of snow/snow after deposition. According to our understanding the analysis method and amount of dust loading on the sample can also alter OC/BC ratio."

Beside this, the OC to EC ratio was also affected by both emission source variability and processing during long-range transport in the atmosphere. EC is a nonvolatile and very stable species, whereas OC contains either many semi volatile species that partition between gas and particle or polar compounds that are preferentially washed out (Granat et al., 2010). So at receptor site the concentration of OC may be less especially during wet seasons.

4. **(a) Related to the point above, how precisely was OC differentiated from BC in the thermal optical technique?**
   **(b) What temperature threshold or thermal evolution profile was applied to separate the two species?**
   **(c) Could this have had anything to do with the high BC/OC ratios that were found in the snow samples?**

Response:
a. There are some uncertainties while differentiating OC from BC in thermal optical techniques. Level of uncertainty depend on amount of dust loading on the sample, temperature protocol, analysis method, and sample type. In our case we adapted IMPROVE protocol (Cao et al., 2003; Chow et al., 2004), and measured the amounts of BC and OC on the quartz filters by using a DRI® Model 2001A thermal optical carbon analyzer. BC/OC ratio may be altered due to below possible reasons.
➤ The different thermal optical methods used to measure OC/BC ratios often produce significantly different results (for same sample) due to variation within the temperature programming and optical techniques followed by each method (Karanasiou et al., 2015).
➤ The OC/BC split point is different for different method and also depend on sample type (residential cook stoves, diesel exhaust, rural aerosols, urban aerosols) (Khan et al., 2011).

➢ Some OC is pyrolytically converted to BC (char) when the sample is heating in inert atmosphere (Zhi et al., 2008).
➢ In thermal optical methods it is hard to avoid the charring of OC and considered as a big challenge to BC and OC measurements (Chow et al., 2004; Schmid et al., 2001).
➢ In general, BC concentrations derived from the IMPROVE method are 1.2–1.5 times higher than those derived from the NIOSH method (Chow et al.,2001; Reisinger et al., 2008), and BC concentrations from the EUSAAR_2 temperature protocol are approximately twice as high as those derived from the NIOSH protocol (Cavalli et al., 2010).

So according to our understanding it may be possible to alter OC/BC ratio by the analysis method, mentioned in lines 313, 315 in the revised manuscript, given below for your reference.

"According to our understanding the analysis method and amount of dust loading on the sample can also alter OC/BC ratio. Further details about OC and BC splitting in thermal optical method are available in Wang el al., 2012".

(b) What temperature threshold was applied to separate the two species?

The IMPROVE_A temperature protocol defines temperature plateaus for thermally derived carbon fractions of
- 120 °C for OC1,
- 250 °C for OC2,           Organic carbon.
- 450 °C for OC3,
- 550 °C for OC4
in a helium (He) carrier gas
Total OC was calculated as OC = OC1+ OC2+ OC3+ OC4. Similarly

- 550 °C for EC1,
- 700 °C for EC2,       Elemental carbon (black carbon).
- 800 °C for EC3
in a 98% He 2% oxygen (O2) carrier gas.
Total EC was calculated as EC = EC1+ EC2+ EC3.

These information are provided in Wang et al., 2012 and we have indicated it in lines 154-155 in the revised manuscript as given below for your reference.

"The temperature threshold that was applied to separate the two species is mentioned in Wang et al., 2012."

(c) Could this have had anything to do with the high BC/OC ratios that were found in the snow samples?

Yes, based on above explanations there may be slight effect on BC/OC ratios. This effect may be more visible in high dust loading samples. We had relatively high dust loading in few samples,

which can affect the BC/OC measurement. We have added related information in lines313-315, quoted below for your reference.

"According to our understanding the analysis method and amount of dust loading on the sample can also alter OC/BC ratios."

> **5. More generally, please describe and if possible quantify, sources of uncertainty in the measurements of BC, OC, and dust in snow.**

Response:

Agreed. We have introduced a separate section to describe the possible sources of uncertainty in the measurements (lines 660-671, given below for your reference).

"The overall precision in the BC, OC and TC concentrations was estimated considering the analytical precision of concentration measurements and mass contributions from field blanks. Uncertainty of the BC and OC mass concentrations was measured through the standard deviation of the field blanks, experimentally determined analytical uncertainty, and projected uncertainty associated with filter extraction. According to our understanding the major uncertainty in our study was the dust effects on BC/OC measurement. Warming role of OC was also not included in the current research, which was low but significant in several regions (Yasunari et al. 2015). Beside this we think snow grain size (snow aging) and snow texture were larger sources of uncertainty in the albedo reduction / radiative forcing calculations than indicated. The measured grain size was usually different from the effective optical grain size used in the SNICAR modeling. Snow grain shape was measured with the help of snow card, but was not used in the online SNICAR albedo simulation model and assumed a spherical shape for the snow grains which may slightly affect the results, because albedo of non-spherical grain is higher than the albedo of spherical grains (Dang et al., 2016)."

> **(6a) My sense is that snow grain size and snow texture are larger sources of uncertainty in the albedo reduction / RF calculations than indicated. Although snow grain size was measured with a hand lens (with reported accuracy of 20um), this determination of grain size is usually different from the effective (surface area-weighted) / optical grain size used in the SNICAR modeling. The true uncertainty in effective/optical grain size is likely much larger than 20um, and I think the paper should include greater acknowledgment of this issue.**
> **The discussion of albedo variability associated with snow grain size (or snow aging) should also more clearly indicate the ranges in snow grain size that were assumed for the albedo modeling.**
>
> **(6b)Furthermore, references to "snow age" are sometimes used when "snow grain size" would be more appropriate, since snow grain size does not always increase monotonically with snow age, and it is really the snow grain size that matters for optical/radiative considerations. Examples of this is are on line 364: "The estimated reduction in snow albedo by dust and BC compounded by the age of snow..." and line 386: "... exact snow age ...".**

**Response:** (6a)

We agreed with the reviewer comment. The discussion of albedo variability associated with snow grain size (or snow aging) has been added in the revised manuscript as suggested (lines 463-474), quoted below for your reference.

"According to our understanding, snow grain size (snow aging) and snow texture were larger sources of uncertainty. The effect of snow grain size is generally larger than the uncertainty in light absorbing particles which varies with the snow type (Schmale et al., 2017). For an effective snow grain radius of 80 μm, 100 μm, 120 μm, the albedo reduction caused by 100 ng g−1 of BC was 0.017, 0.019 and 0.021 respectively. As snow grain size was measured with a hand lens (with reported accuracy of 20 μm), so at least 0.002 uncertainty is present in our albedo results. Snow grain shape was measured with the help of snow card, however grain shape was not used in the online SNICAR albedo simulation model and assumed a spherical shape for the snow grains. Albedo of non-spherical grain is higher than the albedo of spherical grains (Dang et al., 2016). The shapes of snow grains and/or ice crystals is significantly changing with snow age and meteorological conditions during and after snowfall (LaChapelle 1969). Besides this, a number of recent studies (e.g., Flanner et al., 2012; Liou et al., 2014; He et al., 2014, 2017) have shown that both snow grain shape and aerosol-snow internal mixing play important roles in snow albedo calculations."

(6b) Agreed. The "snow age" were removed in the identified locations, lines 419 and 435.
Similarly we made necessary changes in few other locations including lines 457 and 435.

6. **(7a) Snow albedo and perturbations to albedo are modeled and used heavily in this study to derive radiative forcing estimates, but no observations of snow or ice albedo are reported. Are there any observations of snow and ice albedo from this region that could be utilized to help verify or support the modeling?**

   **(7b) I worry in particular that debris could strongly reduce albedo of the glaciers but is neglected in the model, potentially leading to bias in the modeled albedo perturbations.**

**Response:**

7(a) Observations based snow or ice albedo were not estimated in current study. According to our knowledge these are the first such albedo measurements to be done from this region.

7(b) Agreed. The debris could strongly reduce albedo of the glaciers, but the albedo estimated in this study were not from the surface of glaciers or debris covered area. Albedo were only estimated for the snow samples collected from the open mountain valleys as indicated in lines 126-127 and 113.

In current study we estimated the snow albedo through SNICAR model only and there is no in-situ albedo observation. In our next coming paper we are using spectrometer to measure in-situ albedo in this region and to compare it with model results and satellite based snow albedo.

**Minor issues:**

**line 211: "... were put in the above equation and got a c extinction..." - grammar issue.**
Response: Corrected, lines 245, given below for your reference.

"Height of individual trajectory points was put in the above equation and got a normalized extinction profile by assuming surface extinction =1".

**line 251: "... with the generally lower deposition on the Gulkin glacier more affected by other factors" - Which factors?**
Response: Other factors has been added in line 295 of revised manuscript. The whole sentence is given below for your reference.

"The marked difference on the Sachin glacier may have reflected the difference in the direction of air, which comes from Iran and Afghanistan in summer and the Bay of Bengal via India in autumn, with the generally lower deposition on the Gulkin glacier more affected by other factors (such as slope aspect of the glacier and status of local emission near the glacier)."

**line 257: "is considering as" -> "considered as"**

Corrected, line 301.

**line 274: "... low OC/BC ratios can result from a reduction in OC, greater contributions from BC enrichment..." - It is unclear to me which processes "reduction in OC" and "BC enrichment" refer to. Could the authors please elaborate on these processes?**

Response: Below are the possible reasons
  ➤ Since BC in snow was less hydrophilic than OC and thus more OC was scavenged with snow melt water as compared to BC. So OC/BC ratios decreased with time during the snow melting season.
  ➤ One most important factor is post-deposition process, melt water can bring dissolved organic carbon away but not for BC. This may be the one possible reason that we are getting more BC than OC in the snow.
  ➤ The reported OC concentrations here from snow and ice samples was representing water insoluble OC (lines 18, 148, 278, 306); because most of the water-soluble OC was not captured by the filter-based method. Including water-soluble OC could dominate the temporal variation of the OC/BC ratio.
  ➤ Higher concentration BC as compared to OC may also indicates greater melt scavenging of OC and decline of the contribution of coal combustion and/or biomass burning to the carbonaceous aerosol emissions in the major contributing source regions.

➢ In general, BC concentrations derived from the IMPROVE method are 1.2–1.5 times higher than those derived from the NIOSH method (Chow et al.,2001; Reisinger et al., 2008), and BC concentrations from the EUSAAR_2 temperature protocol are approximately twice as high as those derived from the NIOSH protocol (Cavalli et al., 2010).

We add an additional text in revised manuscript lines 307-315, quoted below for your reference.

"One important factor was post-deposition process, melt water can bring dissolved organic carbon away but not for BC. Low OC/BC ratio may also possible due to the fact that OC and BC had redistributed primarily under the control of strong melt water rather than sublimation and/or dry/wet deposition. The OC vs BC correlation in snow and ice samples depend on OC vs BC ratio/concentrations in the atmosphere, post deposition process and then scavenging, enrichment and melt rate of snow/snow after deposition. According to our understanding the analysis method and amount of dust loading on the sample can also alter OC/BC ratios."

**line 329: "albedo of samples from the two sites simulated at a wavelength of 0.975 um ... " - Why are 0.975 um albedo values reported here? Light-absorbing impurities exert the strongest influence on blue or mid-visible albedo (e.g., _0.450 um). The 0.975 um albedo is affected less strongly by impurities, and somewhat heavily by snow grain size, so it seems an odd choice of wavelength to use for reporting albedos.**

Response: Yes, the reviewer is absolutely right. Apology for using a fixed particular wavelength in previous version of manuscript. The sentence has been modified lines 397 in revised manuscript, given below for your reference.

"The values for average albedo of samples from the two sites simulated for MAC values of 7.5, 11, and 15 m$^2$/g and SZA of 57.0–88.9° (day time) under a clear sky ranged from 0.39 (site S1, BC only, midday, MAC 15 m$^2$/g) to 0.85 (site S6, dust only, early evening, MAC 7.5–15 m$^2$/g)."

**line 336: "The results suggest that BC was the dominant forcing factor, rather than dust, as a result of the rapid snowmelt." - The identification of "rapid snowmelt" as the cause for greater BC forcing than dust forcing is confusing here. Perhaps the sentence just needs re-working. Otherwise, what role does snowmelt play in the determination of instantaneous radiative forcing?**

Response: Agreed. The sentence has been modified lines 406-407, as given below for your reference.

"The results suggest that BC was the dominant forcing factor, rather than dust, which influence glacial surface albedo and accelerate glacier melt."

**line 343: "... reduction in daily mean albedo of 1.8 to 2.9% ... " Are these relative or absolute reductions in albedo? If the latter, please use absolute (non-percentage) units. This also applies to other references to percent albedo reduction in the paper.**

**Response:** The albedo reduction values presented here are relative, indicating the difference of albedo with having certain pollutants (BC, or dust, or both) and a reference albedo (with zero pollutants i.e. zero BC and zero dust concentration). Some related text has been added in lines (399 - 401), given below for your reference.

"The albedo reduction values presented here are relative, indicating the difference of albedo with having certain pollutants (BC or dust or both) and a reference albedo (with zero pollutants i.e. zero BC and zero dust concentration)."

**lines 400-401: Which environments do these RF estimates apply to?**

**Response:** Environment and small descript of each reference (used in above mentioned line 400-401) is given below

**Zhang et al. 2017:**
- Study region: Keqikaer Glacier (39°N–46°N and 69°E–95°E) in western Tien Shan.
- Environment: Mid-latitude winter, clearsky, cloudy, cloud amount<5 and for ≥5
- Time period:  May 2015.
- Model used:  SNICAR model (Flanner et al., 2007)
- Radiative forcing:  Obtained by equation used in Kaspari et al., 2014; Yang et al., 2015.

**Nair et al., 2013:**
- Study region: Selected sites/stations in Himalayas region.
- Environment: mid-latitude winter atmospheric conditions.
- Time period: 2005-2011 mainly in pre-monsoon and winter seasons.
- Model used: SNICAR model (Flanner et al., 2007).
- Radiative forcing: Using the short-wave fluxes simulated by SBDART model.

**Yang et al., 2015:**
- Study region: Muji glacier (39.19° N, 73.74° E) in Tibetan Plateau.
- Environment: Clear-sky and cloudy conditions.
- Time period: During snowmelt season of 2012.
- Model used: SNICAR model (Flanner et al., 2007).
- Radiative forcing: SBDART model.

We have added further information in the revised manuscript lines 486-489, quoted below for your reference.

"To estimate these radiative forcing measurements, mid-latitude winter with clear sky and cloudy environment was used by Zhang et al. 2017; mid-latitude winter atmospheric conditions was used by Nair et al., 2013; while clear-sky and cloudy conditions environment was used by Yang et al., 2015."

**lines 406-410: It should be acknowledged again that dust forcing varies strongly with dust optical properties and particle size distribution. The estimates derived here appear to have**

**utilized a generic representation of dust in the model that may or may no be appropriate for the dust that was actually measured.**

Response: Agreed. Below sentences has been added (lines 497-502) in the revised manuscript.

"It is important to mention here that dust forcing varies strongly with dust optical properties, source material and particle size distribution. Properties for dust are unique for each of four size bins used in SNICAR online model. These size bins represent partitions of a lognormal size distribution. We used the estimated size of dust particles with generic property of dust in the model. Some dust particles can have a larger impact on snow albedo than the dust applied here (e.g., Aoki et al., 2006; Painter et al., 2007)."

**lines 461: "BC from East Asia can potentially be lifted up high and transported to the northeast during the summer monsoon season. Nonetheless..." - But transport of East Asia emissions to the northeast does not seem relevant for deposition in Pakistan.**
**Please clarify the relevance of this statement.**

Response: The sentence has been deleted and the paragraph has been slightly modified lines 561-564, given below for your reference.

"The results indicate that only a low level of pollutants (minor contribution) reached the study area from Northwest China. BC particles emitted from distant low latitude source regions such as tropical Africa barely reach the Tibetan Plateau and Himalayan regions because their emissions are removed along the transport pathways during the summer monsoon season (Zhang et al., 2015)."

**line 463: "... low latitude source regions such as South Africa..." - I suggest using "tropical Africa" or something similar here instead of "South Africa" (which happens to also be the name of a country).**
Response: Agreed. Tropical Africa has been used, as suggested (line 563), given below for your reference.

"BC particles emitted from distant low latitude source regions such as tropical Africa barely reach the Tibetan Plateau and Himalayan regions because their emissions are removed along the transport pathways during the summer monsoon season (Zhang et al., 2015)."

**line 464: "weak emissions" - Actually, biomass burning emissions from tropical regions of Africa constitute a substantial share of global BC emissions, so "weak" may not be the best word here.**
Response: Agreed. The sentence has been modified line 564, given below for your reference.

"BC particles emitted from distant low latitude source regions such as tropical Africa barely reach the Tibetan Plateau and Himalayan regions because their emissions are removed along the transport pathways during the summer monsoon season (Zhang et al., 2015)."

**line 479: "considerable" -> "considerably"**

Response: Corrected, lines 578.

**line 482: "The concentration of hydrophobic BC, hydrophilic BC, ... " - The description earlier in Methods indicated only that CO tracers were used. Was BC also simulated with this model? If so, were BC tags applied? Please include more description of the BC simulation in Methods. This seems much more relevant for source attribution, since the physics and chemistry of removal for BC and CO are quite different from each other.**

Response: BC was not simulated with the model.
The purpose of showing concentration of hydrophobic BC (BC1), hydrophilic BC (BC2) was to compare the concentration of fresh (hydrophobic) and aged (hydrophilic) BC during summer and winter seasons over the study region. In this study we applied only CO tracer and it is mentioned in lines 567-568.
We agreed that BC model simulation is relatively more relevant for source attribution. For this time we have CO tracer data (which has relatively good correlation with BC tracer in dry seasons-used by multiple authors in the past Shindell et al., 2008; Chen et al., 2009).
We will use high resolution BC tracer in our next publications in near future. Expected uncertainty in CO tag and some recommendations are stated in lines 672-680 in the revised manuscript. The indicated sentences given below for your reference.

"Future study (BC tracer) will evaluate the details of the different source region of BC reaching the glaciers as compared to region tagged CO tracers."

And

"Better-constrained measurements are required in the future for more robust results. High resolution satellite imagery, high resolution models and continuous monitoring can help us to reduce the present uncertainty."

**line 520: "... and increased grain size and density." - It is not clear to me how snow grain size and snow density should affect the \*concentration\* of BC, as indicated in this sentence. Please clarify.**

Response: Agreed. We removed this portion from the revised sentence (Line 621), the modified sentence given below for your reference.

"The samples from Sost contained the highest average concentration of BC in mountain valleys snow (winter) and those from Kalam the lowest, probably due to the impact of snow age, increased concentration of black carbon and dust (the Sost samples were aged snow and Kalam samples fresh snow)."

**Figure 2: Most of the figure is white space. I suggest shrinking the y-axis range to show the plot values more clearly.**
Response: Agreed. The figure has been modified as suggested, given below for your reference

[Figure]

**Revised Figure 2.**

**Figure 6: Please specify which emission inventory was used and how many days of back-trajectory were simulated.**

Response: Agreed. Emission inventory has been used with number of days as given below for your reference,

[Figure]

Possible polluted regions during summer : South west Russia; western part of Kazakhstan, Iran and Turkmenistan; Uzbekistan; Georgia; north east side of Turkey; Syria; Cyprus.

Possible polluted regions during winter : Iran; Pakistan; Iraq; Turkmenistan; Azerbaijan; Georgia; Jordan; Syria; Tunisia; Ukraine; Libya; Egypt; eastern part of Turkey and south western part of Russia.

**Figure 6. Source contribution regions of pollutants identified using an emissions inventory (Representative Concentration Pathways) coupled with back trajectories (a. 77 simulated days, b. 63 simulated days). Red star indicates the position of the study area.**

**Table 2: Are these relative or absolute snow albedo reductions?**

Response: These albedos are relatives because these albedos were estimated with/from reference albedos (with no dust and no BC in the sample). Some general explanation regarding to how we estimate the albedo is given below.

Albedo were estimated using SNICAR online model by providing input parameters mentioned in Table S1. Model was run four times for one particular sample

1. No dust and no BC (reference albedo),
2. Only dust and no BC,
3. Only BC and no dust,
4. with both dust and BC concentration)

We subtract the albedos obtained in other three options with dust and/or BC from this reference albedo.

We have added these information in the revised manuscript indicated in lines 399-401, given below for your reference

"The albedo reduction values presented here are relative, indicating the difference of albedo with having certain pollutants (BC or dust or both BC and dust) and a reference albedo (with zero pollutants i.e. zero BC and zero dust concentration)."

References:

Aoki, T., Kuchiki, K., Niwano, M., Kodama, Y., Hosaka, M. and Tanaka, T.: Physically based snow albedo model for calculating broadband albedos and the solar heating profile in snowpack for general circulation models, J. Geophys. Res. Atmos., 116(11), 1–22, doi:10.1029/2010JD015507, 2011.

Cao, J. J., Lee, S. C., Ho, K. F., Zhang, X. Y., Zou, S. C., Fung, K., Chow, J. C., and Watson, J. G.: Characteristics of carbonaceous aerosol in Pearl River Delta Region, China during 2001 winter period, Atmos. Environ., 37, 1451-1460, 2003.

Cavalli, F., Viana, M., Yttri, K. E., Genberg, J., and Putaud, J. P.: Toward a standardised thermal-optical protocol for measuring atmospheric organic and elemental carbon: the EUSAAR protocol, Atmos. Meas. Tech., 3, 79–89, https://doi.org/10.5194/amt-3-79-2010, 2010.

Chen, D., Wang, Y., Mcelroy, M. B., He, K., Yantosca, R. M. and Sager, P. Le: and Physics Regional CO pollution and export in China simulated by the high-resolution nested-grid GEOS-Chem model, , (2008), 3825–3839, 2009.

Chow, J. C., Watson, J. G., Crow, D., Lowenthal, D. H., and Merrifield, T.: Comparison of IMPROVE and NIOSH carbon measurements, Aerosol Sci. Tech., 34, 23–34, https://doi.org/10.1080/027868201300081923, 2001.

Chow, J. C., Watson, J. G., Chen, L. W., Arnott, W. P., Moosmüller, H., & Fung, K. (2004). Equivalence of elemental carbon by thermal/optical reflectance and transmittance with different temperature protocols. Environmental Science & Technology, 38, 4414–4422

Cao, J.,Wu, F., Chow, J. C., Lee, S. C., Li, Y., Chen, S.W., An, Z. S., Fung, K. K.,Watson, J. G., Zhu, C. S., and Liu, S. X.: Characterization and source apportionment of atmospheric organic and elemental carbon during fall and winter of 2003 in Xi'an, China, Atmos.Chem. Phys., 5, 3127–3137, doi:10.5194/acp-5-3127-2005,2005.

Dang, C., et al. (2016), Effect of Snow Grain Shape on Snow Albedo. J. Atmos. Sci., 73, 3573–3583, doi:10.1175/JAS-D-15-0276.1

Ducret, J. and Cachier, H.: Particulate carbon content in rain at various temperate and tropical locations, J. Atmos. Chem., 15, 55–67, 1992.

Fitzgerald, W. F.: Clean hands, dirty hands: Clair Patterson and the aquatic biogeochemistry of mercury, Clean Hands, Clair Patterson's Crusade Against Environmental Lead Contamination, 119–137, 1999.

Flanner, M. G., et al. (2012), Enhanced solar energy absorption by internally-mixed black carbon in snow grains, Atmos. Chem. Phys., 12, 4699-4721, doi:10.5194/acp- 12-4699-2012.

Ganguly, D., Rasch, P. J., Wang, H. and Yoon, J.: Climate response of the South Asian monsoon system to anthropogenic aerosols, JGR-D, 117(May), 1–20, doi:10.1029/2012JD017508, 2012.

Granat, L., J. E. Engström, S. Praveen, and H. Rodhe (2010), Light absorbing material (soot) in rain water and in aerosol particles in the Maldives, J. Geophys. Res., 115, D16307, doi:10.1029/2009JD013768

He, C., et al. (2014), Black carbon radiative forcing over the Tibetan Plateau, Geophys. Res. Lett., 41, 7806–7813, doi:10.1002/2014GL062191.

He, C., et al. (2017): Impact of Snow Grain Shape and Black Carbon-Snow Internal Mixing on Snow Optical Properties: Parameterizations for Climate Models. J. Climate, 0, doi:10.1175/JCLI-D-17-0300.1

Karanasiou, A., Minguillón, M. C., Viana, M., Alastuey, A., Putaud, J. P., Maenhaut, W., Panteliadis, P., Moˇcnik, G., Favez, O., and Kuhlbusch, T. A. J.: Thermal-optical analysis for the measurement of elemental carbon (EC) and organic carbon (OC) in ambient air a literature review, Atmos. Meas. Tech. Discuss.,8, 9649–9712, https://doi.org/10.5194/amtd-8-9649-2015, 2015.

Kaspari, S., Painter, T. H., Gysel, M., Skiles, S. M. and Schwikowski, M.: Seasonal and elevational variations of black carbon and dust in snow and ice in the Solu-Khumbu, Nepal and estimated radiative forcings, Atmos. Chem. Phys., 14(15), 8089–8103, doi:10.5194/acp-14-8089-2014, 2014.

Khan, B., Hays, M. D., Geron, C., Jetter, J., Khan, B., Hays, M. D., Geron, C., Jetter, J., Khan, B., Hays, M. D., Geron, C. and Jetter, J.: Differences in the OC / EC Ratios that Characterize Ambient and Source Aerosols due to Thermal- Optical Analysis Differences in the OC / EC Ratios that Characterize Ambient and Source Aerosols due to Thermal-Optical Analysis, , 6826(October 2017), doi:10.1080/02786826.2011.609194, 2012.

Kuhlmann, J. and Quaas, J.: and Physics How can aerosols affect the Asian summer monsoon ? Assessment during three consecutive pre-monsoon seasons from CALIPSO satellite data, , 1930, 4673–4688, doi:10.5194/acp-10-4673-2010, 2010.

Kroll J Het al 2011 Carbon oxidation state as a metric for describing the chemistry of atmospheric organic aerosol Nature Chem. 3 133–9

LaChapelle, E. R., 1969: Field Guide to Snow Crystals. University of Washington Press, 112 pp.

Lamarque, J.F., Bond, T.C., Eyring, V., Granier, C., Heil, A., Klimont, Z., Lee, D., Liousse, C., Mieville, A., Owen, B., Schultz, M.G., Shindell, D., Smith, S.J., Stehfest, E., Van Aardenne, J., Cooper, O.R., Kainuma, M., Mahowald, N., McConnell, J.R., Naik, V., Riahi, K., Van Vuuren, D.P., 2010. Historical (1850-2000) gridded anthropogenic and biomass burning emissions of reactive gases and aerosols: Methodology and application. Atmospheric Chemistry and Physics 10, 7017–7039, 2010.

Li, X., Kang, S., He, X., Qu, B., Tripathee, L., Jing, Z., Paudyal, R., Li, Y., Zhang, Y., Yan, F., Li, G. and Li, C.: Light-absorbing impurities accelerate glacier melt in the Central Tibetan Plateau, Sci. Total Environ., doi:10.1016/j.scitotenv.2017.02.169, 2017.

Li, Y., Chen, J., Kang, S., Li, C., Qu, B., Tripathee, L., Yan, F., Zhang, Y., Guo, J., Gul, C. and Qin, X.: Impacts of black carbon and mineral dust on radiative forcing and glacier melting during summer in the Qilian Mountains, northeastern Tibetan Plateau, Cryosph. Discuss., (April), 1–14, doi:10.5194/tc-2016-32, 2016.

Liou, K. N., et al. (2014), Stochastic parameterization for light absorption by internally mixed BC/dust in snow grains for application to climate models, J. Geophys. Res. Atmos., 119, doi:10.1002/2014JD021665.

Ming, J., Xiao, C., Cachier, H., Qin, D., Qin, X., Li, Z. and Pu, J.: Black Carbon (BC) in the snow of glaciers in west China and its potential effects on albedos, Atmos. Res., 92(1), 114–123, doi:10.1016/j.atmosres.2008.09.007, 2009.

Nair, V. S., Babu, S. S., Moorthy, K. K., Sharma, A. K., Marinoni, A. and Ajai: Black carbon aerosols over the Himalayas: Direct and surface albedo forcing, Tellus, Ser. B Chem. Phys. Meteorol., 65(1), doi:10.3402/tellusb.v65i0.19738, 2013.

Niu, H., Kang, S., Shi, X., Paudyal, R., He, Y., Li, G. and Wang, S.: Science of the Total Environment In-situ measurements of light-absorbing impurities in snow of glacier on Mt . Yulong and implications for radiative forcing estimates, Sci. Total Environ., 581–582, 848–856, doi:10.1016/j.scitotenv.2017.01.032, 2017.

Novakov, T., Menon, S., Kirchstetter, T. W., Koch, D. and Hansen, J. E.: Aerosol organic carbon to black carbon ratios : Analysis of published data and implications for climate forcing of soot emissions maybe a useful approach to slow global warming ., , 110, 1–13, doi:10.1029/2005JD005977, 2005.

Ohara, T. et al., 2007, An Asian emission inventory of anthropogenic emission sources for the period 1980-2020. Atmospheric Chemistry and Physics 7, 4419-4444

Painter, T. H., Barrett, A. P., Landry, C. C., Neff, J. C., Cassidy, M. P., Lawrence, C. R., McBride, K. E. and Farmer, G. L.: Impact of disturbed desert soils on duration of mountain snow cover, Geophys. Res. Lett., 34(12), 1–6, doi:10.1029/2007GL030284, 2007.

Qian, Y., Flanner, M. G., Leung, L. R. and Wang, W. 2011. Sensitivity studies on the impacts of Tibetan Plateau snowpack pollution on the Asian hydrological cycle and monsoon climate. Atmos. Chem. Phys. 11, 1929-1948.

Qu, B., Ming, J., Kang, S. C., Zhang, G. S., Li, Y. W., Li, C. D., Zhao, S. Y., Ji, Z. M. and Cao, J. J.: The decreasing albedo of the Zhadang glacier on western Nyainqentanglha and the role of light-absorbing impurities, Atmos. Chem. Phys., 14(20), 11117–11128, doi:10.5194/acp-14-11117-2014, 2014.

Ram Kand SarinMM2010 Spatio-temporal variability in atmospheric abundances of EC,OC and WSOC over Northern India J. Aerosol Sci. 41 88–98

Reisinger, P., Wonaschütz, A., Hitzenberger, R., Petzold, A., Bauer, H., Jankowski, N., Puxbaum, H., Chi, X., and Maenhaut, W.: Intercomparison of measurement techniques for black or elemental carbon under urban background conditions in wintertime: influence of biomass combustion, Environ. Sci. Technol., 42, 884–889, https://doi.org/10.1021/es0715041, 2008.

Saarikoski S et al 2008 Sources of organic carbon in fine particulate matter in Northern European urban air Atmos. Chem. Phys. 8 6281–95

Schmid, H., Laskus, L., Abraham, H. J., Baltensperger, U., Lavanchy, V., Bizjak, M., et al. (2001). Results of the "carbon conference" international aerosol carbon round robin test stage I. Atmospheric Environment, 35, 2111–2121.

Schmale, J., Flanner, M., Kang, S., Sprenger, M., Zhang, Q., Guo, J., Li, Y., Schwikowski, M. and Farinotti, D.: Modulation of snow reflectance and snowmelt from Central Asian glaciers by anthropogenic black carbon, Sci. Rep., 7(October 2016), 40501, doi:10.1038/srep40501, 2017.

Shindell, D. T., Chin, M., Dentener, F.: A multi-model assessment of pollution transport to the Arctic, Atmos. Chem. Phys., 8, 5353–5372, 2008, http://www.atmos-chem-phys.net/8/5353/2008/.

Wang M, B Xu, J Cao, X Tie, H Wang, R Zhang, Y Qian, PJ Rasch, S Zhao, G Wu, H Zhao, DR Joswiak, J Li, and Y Xie. 2015. "Carbonaceous Aerosols Recorded in a Southeastern Tibetan Glacier: Analysis of Temporal Variations and Model Estimates of Sources and Radiative Forcing." Atmospheric Chemistry and Physics 15:1191-1204. doi:10.5194/acp-15-1191-2015.

Wang, X., Pu, W., Ren, Y., Zhang, X., Zhang, X., Shi, J., Jin, H., Dai, M. and Chen, Q.: Snow albedo reduction in seasonal snow due to anthropogenic dust and carbonaceous aerosols across northern China, Atmos. Chem. Phys. Discuss., (September), 1–52, doi:10.5194/acp-2016-667, 2016.

Wang, X., Pu, W., Ren, Y., Zhang, X., Zhang, X., Shi, J., Jin, H., Dai, M. and Chen, Q.: Snow albedo reduction in seasonal snow due to anthropogenic dust and carbonaceous aerosols across northern China, Atmos. Chem. Phys. Discuss., (September), 1–52, doi:10.5194/acp-2016-667, 2016.

Wang M, B Xu, J Cao, X Tie, H Wang, R Zhang, Y Qian, PJ Rasch, S Zhao, G Wu, H Zhao, DR Joswiak, J Li, and Y Xie. 2015. "Carbonaceous Aerosols Recorded in a Southeastern Tibetan Glacier: Analysis of Temporal Variations and Model Estimates of Sources and Radiative Forcing." Atmospheric Chemistry and Physics 15:1191-1204. doi:10.5194/acp-15-1191-2015.

Xu, B., Yao, T., Liu, X. and Wang, N.: Elemental and organic carbon measurements with a two-step heating-gas chromatography system in snow samples from the Tibetan Plateau, Ann. Glaciol., 43(June 2015), 257–262, doi:10.3189/172756406781812122, 2006.

Xu, B., Cao, J., Joswiak, D. R., Liu, X., Zhao, H. and He, J.: Post-depositional enrichment of black soot in snow-pack and accelerated melting of Tibetan glaciers, Environ. Res. Lett., 7(1), 14022, doi:10.1088/1748-9326/7/1/014022, 2012.

Yamaji, K. et al., 2003, A country-specific, high-resolution emission inventory for methane from livestock in Asia in 2000. Atmospheric Environment 37(31), 4393-4406, doi:10.1016/S1352-2310(03)00586-7

Yan X. et al., 2003, Development of region-specific emission factors and estimation of methane emission from rice field in East, Southeast and south Asian countries, Global Change Biology, 9, 237-254.

Yang, S., Xu, B., Cao, J., Zender, C. S. and Wang, M.: Climate effect of black carbon aerosol in a Tibetan Plateau glacier, Atmos. Environ., 111, 71–78, doi:10.1016/j.atmosenv.2015.03.016, 2015.

Zhang R, H Wang, Y Qian, PJ Rasch, RC Easter, Jr, PL Ma, B Singh, J Huang, and Q Fu. 2015. "Quantifying sources, transport, deposition, and radiative forcing of black carbon over the Himalayas and Tibetan Plateau." Atmospheric Chemistry and Physics 15(11):6205-6223. doi:10.5194/acp-15-6205-2015.

Zhang, Y., Kang, S., Xu, M., Sprenger, M., Gao, T., Cong, Z., Li, C., Guo, J., Xu, Z., Li, Y., Li, G., Li, X., Liu, Y. and Han, H.: Sciences in Cold and Arid Regions Light-absorbing impurities on Keqikaer Glacier in western Tien Shan : concentrations and potential impact on albedo reduction, , 9(2), doi:10.3724/SP.J.1226.2017.00097.Light-absorbing, 2017.

Zhang, Y., Hirabayashi, Y., Liu, Q. and Liu, S.: Glacier runoff and its impact in a highly glacierized catchment in the southeastern Tibetan Plateau: Past and future trends, J. Glaciol., 61(228), 713–730, doi:10.3189/2015JoG14J188, 2015.

Zhi, G.: Effects of temperature parameters on thermal- optical analysis of organic and elemental carbon in aerosol, , (November 2014), doi:10.1007/s10661-008-0393-4, 2008.

Zhang, Y., Hirabayashi, Y., Liu, Q. and Liu, S.: Glacier runoff and its impact in a highly glacierized catchment in the southeastern Tibetan Plateau: Past and future trends, J. Glaciol., 61(228), 713–730, doi:10.3189/2015JoG14J188,

2015.

**Thank you**

---

## Author Comment (AC2) · 10 Jan 2018

**Short comment by Cenlin HE**

a) I have a minor comment related to the snow albedo calculations including aerosol contamination. The authors used the SNICAR model to calculate snow albedo contaminated by aerosols. If I understand correctly, the authors assumed external mixing of snow and aerosols as well as spherical snow grains. I suggest that the authors explicitly state their assumptions here.

b) Besides, a number of recent studies (e.g., Flanner et al., 2012; Liou et al., 2014; Dang et al., 2016; He et al., 2014, 2017) have shown that both snow grain shape (nonspherical vs. spherical) and aerosol-snow internal mixing play important roles in snow albedo calculations. Particularly, non-spherical snow grains reduces snow albedo reductions caused by light-absorbing aerosols compared with spherical snow grains, while aerosol-snow internal mixing significantly enhances snow albedo reductions compared with external mixing. It will be helpful if the authors could include these recent studies and add some discussions on this aspect.

Response:

a) Thank for a minor but valuable comment on our manuscript. Yes we explicitly stated the assumptions in the revised manuscript (line number 138, 464-475).

b) The mentioned references are really interested, indicating role of snow grain shape and mixing of aerosol with snow. Thank you to provide us the related references and we have added some discussion on the basis of these references and cited all the provided references.

[Figure]
 Thank you

---

## Author Comment (AC3) · 10 Jan 2018

**This manuscript reported the data of light absorbing particles (LAP, such BC, OC and dust) measured in snow/ice in northern Pakistan and estimated the induced snow albedo reduction and corresponded radiative forcing during 2015-2016. Authors found the concentration of BC, OC and dust in aged snow is higher than in fresh snow and ice and the concentration over northern Pakistan is higher than over the Himalayas and Tibetan Plateau. Estimated LAP-induced daily mean snow albedo reduction is approximately 0.07-12.0% and corresponded radiative forcing is approximately 0.16-43.5 Wm-2, depending on snow type, solar zenith angle, and locations. Also different methods are used in this study to identify the source regions of pollutants measured in this region.**

**Overall the results are interesting and measurement data are valuable for the community. The figures and tables in the manuscript are relevant, but not very good in quality, and need to be improved. In general the paper is well written but in many places the English could and should be improved. There are some major weaknesses in the manuscript, especially in the source region identification part (3.5) and aerosol type frequency distribution part (3.2). After the below comments are appropriately addressed, I would suggest to accept the manuscript for publication in the Atmospheric Chemistry and Physics.**

Response:
We thank the reviewer for their comments that significantly contributed to improving the original manuscript. Please see below our comment-by comment-responses to each of the reviewer's comments and suggestions.

Reviewer Comments in black

Responses in blue

Modified text in the revised manuscript is in green.

**Technical comments:**

**Introduction, I would suggest use Light-absorbing particles (LAP) instead of Light absorbing impurities, see Qian et al., 2015, which by the way is a review article for both measurement methods and modeling activities. This article is so relevant so probably should be cited in Introduction part.**

Response:

Light-absorbing particles (LAP) has been used instead of Light absorbing impurities (lines 14,48,82,89,151,292,418,420, and 453). We changed the title accordingly. The recommended article has been cited in the introduction (lines 83). The text has been quoted below for your reference.

"A number of authors have described the concentration and impacts of light absorbing particles in the Tibetan glaciers (for example Qian et al., 2015; Wang et al., 2015; Que et al., 2016; Zhang et al., 2017; Li et al., 2017; Niu et al., 2017)."

**Lines 62-65, Besides warming efficiency, another important characteristic of LAPs is its higher snowmelt efficiency, see Qian et al., 2011.**

Response:
An additional sentence related to warming has been added in the revised manuscript (lines 67-69).

"Besides warming efficiency, another important characteristic of BC is its higher snowmelt efficiency. The snowmelt efficacy induced by BC in snow is larger for snow cover fraction and snow water equivalent than induced by carbon dioxide increase (Qian et al., 2011)."

**Lines 110, precipitation 0.412 +/- 2 mm for per day or per year?**

Response:
The sentences have been modified (lines 117-119), given below for your reference.

"According to the 10 years record (1999–2008) of the two nearby climatic stations, the mean total annual precipitation was 170 mm at Khunjerab (36.83°N, 75.40°E, 4730 m) station, and 680 mm at Naltar (36.29°N, 74.12°E,2858 m) station,"

**Lines 123-128, again sample method is summarized in Qian et al., 2015.**

Response:
We have referred to Qian et al., 2015 in the methodology section (line 138).

"Qian et al., 2015 summarized sample methods for light absorbing particles in snow and ice from different region including Arctic, Tibetan Plateau and mid-latitude regions."

**Section 2.3.1, without 2.3.2, really needs 2.3.1? More details are needed regarding how the L-2 data are processed.**

Response:
As per reviewer suggestions, the title/ heading of section 2.3.1 has been deleted (line 162). This part (paragraph) describes the aerosol subtypes, and CALIPSO level 2 lidar data processing.

We have also added details about L-2 data processing in the modified manuscript as quoted below (lines 163, 169-174).

"The CALIPSO models define aerosol subtypes, with 532-nm (1064 nm) extinction-to-backscatter ratio. The CALIPSO Level 2 lidar vertical feature mask data product describes the vertical and horizontal distribution of clouds and aerosol layers (downloaded from https://eosweb.larc.nasa.gov/project/calipso/aerosol_profile_table). On the basis of observed backscatter strength and depolarization, the aerosol subtypes have been pre classified in the downloaded data. The details of algorithm used for the classification have been presented in Omar et al., 2009. Percentage contribution of individual aerosol subtypes were plotted using Matlab."

The number of counts for a specific aerosol type in specific month were plotted as indicated in Figure 2 and Figure S4. According to our understanding, few authors in the past (including Cong et al., 2015; Ali H. Omar et al., 2009 and Wang et al., 2016) used the sub-type aerosol data.

**Lines 155-159, This paragraph should be removed or moved to Introduction section.**

Response:
The paragraph has been moved to introduction section with few modifications as suggested by the reviewer (lines 47-49 of modified manuscript). The whole sentence is given below for your reference.

"However, the exact amount of albedo reduction also depends on the refractive index, snow age, grain size, solar zenith angle (SZA), snow density, dust particle size and concentration, particle morphology, surface roughness, snow depth, liquid water content, snow shape and topography (Wiscombe and Warren 1985)."

**Section 2.5 is very poorly organized and kind of just present whatever tools you have or used before, without a clear goal or coherence in science structure. Must be rewritten.**

Response:
        The section has been reorganized and edited by a native English speaking editor (lines 210-269). The modified section has been quoted below for your reference.

[revised manuscript text omitted]

**2.5.1 Wind maps, why not use 2005-2006 wind maps instead of 50-year average? 700mb is very high level for low-elevation region and aerosol concentration is very low at that high level. I would suggest use the terrain-oriented level like sigma level near the surface. I would also strongly suggest (a must do) use MERRA-2 reanalysis data, in which not only the data quality is better than NCEP/NCAR but also it includes aerosol data that can be used to compare with the measurement and is more appropriate for looking at the long-distance transport. Suggestion**

Response:

Our sampling time period was 2015-2016 so we believe the reviewer wanted us to plot wind maps for 2015-2016.

As per your suggestions, we have used the MERRA-2 reanalysis data at 850mb and replotted the wind maps, during the selected months of 2015-2016 as shown for your reference below. The same figure has been used in the revised manuscript.

[Figure]

**Figure 5. Monthly average horizontal wind patterns at 850 hPa during a) May, b) June, c) December, and d) January, corresponding to approximately 2500 masl, from GES DISC. The study area is indicated by a star, and white lines indicating streamlines. The background colors show monthly mean aerosol optical depth.**

**2.5.3,**
  **a. WRF-STEM can only tag CO, because of many differences between CO and LAP in such as emission sources, chemistry and removal, how to quantify their differences in long-range transport and source identifications?**
  **b. How to infer the transport and source for LAPs based on CO and what's the uncertainty?**
  **Please see Zhang et al. 2015 and Wang et al., 2015 for source detection methods used over Tibetan Plateau region.**

Response:

a.  We agree with the reviewer comment, however for the time being we have CO tracer data (which has relatively good correlation with BC tracer in dry seasons) and high resolution BC tracer will be used in our next publication in near future (indicated in lines 587 and 677-680).

➢ BC and CO both are primary pollutants and emit from almost similar anthropogenic sources (may be different in concentration). We were more precise on the source region identification of these pollutants.

➢ Region tagged CO tracer is a standard air quality modeling tool used by other regional and global chemical transport models to identify pollution source regions (Chen et al., 2009; Park et al., 2009; Lamarque and Hess, 2003) lines 249-251. The WRF-STEM model uses region tagged carbon monoxide (CO) tracers for many regions in the world to identify geographical areas contributing to observed pollutants (Adhikary et al., 2010), lines 252.

➢ Each model has some strength and some weaknesses. The strength of our applied model is its relatively higher spatial resolution (line 253). For rough and complex topography of mountain region, it is important to use high resolution model.

b. Transport, source of LAPs based on CO and uncertainty
   Source
        ➢ Depending on event, the source of LAPs (BC) and CO may be different, but in several cases the source of BC and CO might be same such as biomass burning (cooking, outdoor burning, and forest fires) and incomplete combustion process.
        ➢ During incomplete combustion of carbon-containing fuels, such as gasoline, natural gas, oil, coal, and wood, emits CO as well as BC.
   Transport
        ➢ Based on chemical reactivity, the weight of the particle and its life time in the atmosphere, the transport of BC and CO may be different.
          During dry seasons, CO and BC have quite similar transport but in wet seasons (especially during monsoon seasons) BC particles are washed out with precipitation and relatively higher concentrations of CO are reaching receptor

sites as compared to BC. Below quoted text has been added in the revised manuscript (lines 674-675).

"In our case we analyze the model from 1st June to 4th July during summer season and 15th December to 17th January during winter season. In Pakistan monsoon is generally starting during the first week of July each year, so we are expecting relatively good correlation in transport between CO and BC during pre-monsoon period. A cool, dry winter starts from December through February each year."

Uncertainty

Related uncertainty is mentioned in lines 671-680, given below for your reference.

"On modeling side the possible uncertainties are related to using CO as a tracer for light absorbing particles source region. Uncertainties are also attributed to errors in emissions inventories, simulated meteorology and removal processes built in the model. The physics and chemistry of removal for BC and CO are different from each other especially during wet seasons. We analyze the model during pre-monsoon and relatively dry periods, so we are expecting relatively good correlation in transport between CO and BC. While using global emission inventories we were unable to capture emissions at local scale. Contributions of local sources may be underestimated by coarse-resolution models. Therefore high resolution models and emission inventories at local scale are required to capture local emissions. Better-constrained measurements are required in the future for more robust results. High resolution satellite imagery, high resolution models and continuous monitoring can help us to reduce the present uncertainty."

Zhang et al. 2015 and Wang et al., 2015 are presenting impressive work for source detection methods in the Tibetan Plateau region. We have cited these papers in the proper locations.

We agreed with the reviewer comment and using high resolution BC tracer WRF-STEM model in our next publication.

**Line 257, 24 hours. Considering–>considered.**
Response:
Corrected (line 301, in the revised manuscript)

**Line 318, Jun–>June.**

Response:
Corrected (line 384, in the revised manuscript)

**Section 3.2, very weak! How to connect the conclusion from this section with other parts?**

Response:

We have rewritten and improved the section according to the suggestions. Repeated sentences have been removed. Some additional text has been added to improve the connection of conclusions from this section to other parts of the paper, lines 379-390.

"The CALIPSO aerosol type identifications analysis indicated that "smoke" was the most frequent-occurring type of aerosol over the study region during both summer and winter seasons. This result indicate that biomass burning sources may be the dominant contributor in this region. Frequency of subtype aerosols for the month of June in 2006 to 2014 is shown in Figure S4. Figure 2 shows the seasonal results for month of May, June (summer) and December, January (winter) in the form of a box plot. During June smoke had the highest frequency (39%), followed by dust (21%), polluted dust (12%), and others (20%) Figure S4. Overall Smoke, dust and or polluted dust were the dominant subtype aerosols in selected months over the study region. This type of aerosol measurement in the atmosphere is important for our current study because it provides observation based data over the study region. Other approaches used (such as modeling) were based on interpolation not observation. Pollutant deposition depends on the concentration of pollutants in the atmosphere, the results are consistent with the high concentration of BC (from smoke) and dust particles in the glacier and snow surface samples."

**Section 4 Summary and conclusion, one more section should be added for discussion in uncertainty and possible future direction for both modeling and measurement campaign. For example, how snow aging (snow grain size) and melting water scavenging efficiency (see Qian et al, 2014) affect the conclusions?**

Response:
Agreed. We have added one more section for discussion on uncertainty and possible future directions in section 4, lines 660-671, given below for your reference.

"The overall precision in the BC, OC and TC concentrations was estimated considering the analytical precision of concentration measurements and mass contributions from field blanks. Uncertainty of the BC and OC mass concentrations was measured through the standard deviation of the field blanks, experimentally determined analytical uncertainty, and projected uncertainty associated with filter extraction. According to our understanding the major uncertainty in our study was the dust effect on BC/OC measurement. Warming role of OC was also not included in the current research which was low but significant in several regions (Yasunari et al. 2015). Beside this we think snow grain size (snow aging) and snow texture were larger sources of uncertainty in the albedo reduction / radiative forcing calculations. The measured grain size was usually different from the effective optical grain size used in the SNICAR modeling. Snow grain shape was measured with the help of snow card, but was not used in the online SNICAR albedo simulation model and assumed a spherical shape for the snow grains which may slightly affect the results, because albedo of non-spherical grain is higher than the albedo of spherical grains (Dang et al., 2016). On modeling side the possible uncertainties are related to using CO as a tracer for light absorbing particles source region. Uncertainties are also attributed to errors in emissions inventories, simulated meteorology and removal processes built in the model. The physics and chemistry of removal for BC and CO are different from each other especially during wet seasons. In order to reduce uncertainty in source region high resolution BC tracer are required. Better-constrained measurements are required in the future for more robust results. High resolution satellite imagery, high resolution models and continuous monitoring can help us to reduce the present uncertainty."

**Figure 2, this is a poor figure and should be re-designed. For example, reduce the y-axis range from 300 to 150. Btw why the numbers for y-axis are 50, 100, 150, 200, 150 (should be 250?)?**

Response:
Figure has been re-designed as suggested.
                    Revised figure 2 is given below for your reference:

[Figure]

**Figure 2. Frequency distribution of aerosol subtypes in the atmosphere over the study region calculated from CALIPSO data for the months in 2006 to 2014.**

**Figure 3, give full name for MAC in figure caption. Also consider use identical range for y-axis e.g. 0.4-1.0 and for x-axis 0.3-1.2 for Panel c and d.**

Response:
Figure 3 has been re-designed as suggested. Full name for MAC has been used, identical range for y-axis e.g. 0.4-1.0 and for x-axis 0.3-1.2 for Panel c and d has been used. Quality of the figure has been improved.

Revised Figure 3 is given below for your reference:

[Figure]

**Figure 3.** Spectral variation in albedo for winter sampling sites and selected Mass Absorption Cross section (MAC) values, (a) average albedo of samples at each of the sites (b) daily mean albedo reduction of fresh snow (site S6) and aged snow (site S1) snow, (note different scales of y axis) (c) albedo of fresh snow site S6, (d) albedo of aged snow site S1.

**Figure 4,**
   a) **Suggest use identical y-axis range so can highlight the bigger effect over aged snow. The unit for radiative forcing is %?**
   b) **More discussion should be provided regarding how snow aging affect the albedo reduction and radiative forcing (e.g. Qian et al., 2014)?**

Response:
   a. Agreed. An identical y-axis range has been used. Radiative forcing mentioned here was calculated from albedo reductions indicated on left side of the figure. % symbol has been used in the caption of this figure. Revised Figure 4 is given below for your reference:

[Figure]

**Figure 4. Daily mean radiative forcing reuction and albedo reduction caused by black carbon and dust, for different Mass Absorption Cross section (MAC) in (a) fresh (low black carbon) and (b) aged (high black carbon) snow samples (note different scales of y axis)**

b. Agreed. More text has been added regarding how snow aging affects the albedo reduction and radiative forcing (line 505-510), given below for your reference

"Snow aging (snow grain size) plays an important role in albedo reduction and radiative forcing. Schmale et al., (2017) stating that the effect of snow grain size is generally larger than the uncertainty in light absorbing particles which varies with the snow type. The impact of snow aging factor on BC in snow and induced forcing are complex and had spatial and seasonal variation (Qian et al., 2014). Increase of snow aging factor reduces snow albedo and accelerate the snow melting."

**Figure 5, what blue contours represent? Again 700 mb is too high and MERRA-2 is a much better dataset.**

**Response:**
The blue lines in previous Figure 5 were indicating streamlines. We used 850 mb in the revised figure by using MERRA-2 reanalysis data, as suggested. Revised figure given below for your reference.

[Figure]

**Figure 5. Monthly average horizontal wind patterns at 850 hPa during a) May, b) June, c) December, and d) January, corresponding to approximately 2500 masl, from GES DISC. The study area is indicated by a star, and white lines indicating streamlines. The background colors show monthly mean aerosol optical depth.**

**Figure 6, not clear what color shades represent?**

**Response:**
Agreed. The confusing color shades have been removed. We modified the figure (line 933), as given below for your reference.

[Figure]

Possible polluted regions during summer : South west Russia; western part of Kazakhstan, Iran and Turkmenistan; Uzbekistan; Georgia; north east side of Turkey; Syria; Cyprus.

Possible polluted regions during winter : Iran; Pakistan; Iraq; Turkmenistan; Azerbaijan; Georgia; Jordan; Syria; Tunisia; Ukraine; Libya; Egypt; eastern part of Turkey and south western part of Russia.

**Figure 7,**
   **(a) I am not sure how the quantitative number of contributions are meaningful because the numbers for LAP could be very different with that for CO.**
   **(b) Anyway again, Section 2.5 is very poorly organized and kind of just present whatever tools you have or used before, without a clear goal or coherence in science structure. Need to be rewritten with a clear conclusion.**

Response:
   (a) Yes, we agreed that the numbers used in the Figure 7 could be different than that for LAP, especially during wet seasons. Based on following justifications, we are expecting relatively less difference between the numbers for LAP and CO.

   ➤ We analyze the model from 1 June to 4 July during the summer season. In Pakistan, monsoon generally starts during the first week of July each year, so we are expecting relatively good correlation in transport between CO and BC during the pre-monsoon period.
   ➤ During winter we analyze the model from 15 December to 17 January. A cool, dry winter starts from December through February each year. Winter season is dry but clouds during this season may bring some uncertainty in our results.

We mentioned this uncertainty in multiple places in the revised manuscript, including lines 672-680. High resolution WRF-STEM BC tagged will be used in our next publication.

   (b) We have reorganized the section 2.5 and made multiple changes. We tried our best to mention a clear goal in this section with a clear conclusion, lines 205-269.

**Table 2, give full name for MAC (in other tables/ figures as well).**

Response:
Agreed. We used the full name for MAC at all necessary locations, such as lines 48, 58, 61 in supplementary document, and line 917 (caption of Figure 3) in the main manuscript.

**Figure S7, give full names for BC1 and BC2.**

Response:
Agreed. Full names for BC1 and BC2 has been used, given below for your reference

[Figure]

**Figure S7. Concentration of black carbon1, black carbon2 and black carbon on the Sachin glacier calculated using the WRF-STEM model: a) summer, b) winter.**

**Suggested references**
    **Qian et. al, 2015.**
    **Zhang et. al, 2015.**
    **Qian et. al, 2014.**
    **Wang et. al, 2015.**
    **Qian et. al, 2011.**

Response:
We thank this anonymous reviewer for his careful and detailed reviews and suggestions that helped us to greatly improve this paper. Beside this the provided references/articles are very relevant and presenting a quality work in the region. So we used/cited properly these articles in the revised manuscript. We feel a big improvement in the text after revising and citing these articles.

[revised manuscript text omitted]

Wang, Mo, et al. "Two distinct patterns of seasonal variation of airborne black carbon over Tibetan Plateau." Science of the Total Environment 573 (2016): 1041-1052.

Yasunari, T. J., Lau, K.-M., Mahanama, S. P. P., Colarco, P. R., Silva, A. M. Da, Aoki, T., Aoki, K., Murao, N., Yamagata, S. and Kodama, Y.: The GOddard SnoW Impurity Module (GOSWIM) for the NASA GEOS-5 Earth System Model: Preliminary Comparisons with Observations in Sapporo, Japan, Sola, 10(MAY), 50–56, doi:10.2151/sola.2014-011, 2014.

Zhang R, H Wang, Y Qian, PJ Rasch, RC Easter, Jr, PL Ma, B Singh, J Huang, and Q Fu. 2015. "Quantifying sources, transport, deposition, and radiative forcing of black carbon over the Himalayas and Tibetan Plateau." Atmospheric Chemistry and Physics 15(11):6205-6223. doi:10.5194/acp-15-6205-2015.

Thank you

---

## Author Response (AR2)

**Response to minor comment on "Concentrations and source regions of light absorbing particles in snow/ice in northern Pakistan and their impact on snow albedo" by Chaman Gul et al.**

We thank the Co-Editor and reviewers for their comments. Please see responses to each of the comments below, coded as follows:
Reviewer comments in black
Responses in blue
Modified text in revised manuscript in green

The authors have provided a comprehensive response to concerns raised by the reviewers on the first submission of this manuscript. Overall, I think the authors have adequately addressed these concerns, though the following issues should be addressed prior to publication:

**1.** The paper needs to be carefully and thoroughly edited for grammar and clarity. Reading only the quoted passages of edited text in the response to reviewer's document, I see numerous instances of incomplete sentences, incorrect grammar, and unclear descriptions.

The present manuscript has been thoroughly edited by a native English speaker editor; this is acknowledged in the Acknowledgements section.

**2.** One question I initially raised was whether the reported albedo reductions are absolute or relative. The authors explained their calculations and describe them as "relative" changes, but I think they are actually "absolute" changes in the context I was inquiring about. I wasn't completely clear on the distinction, however, so I will clarify here what I meant. Let's say two albedo measurements are 0.70 and 0.80. The absolute difference between these could be reported as 0.10 or 10%. The relative change could be reported as: (0.80-0.70)/0.70 = 14%. Hence when I see "albedo change" reported with a "%", I don't know if it is the absolute change or relative change. I believe the authors are reporting absolute change and if so they should clarify this. Simply reporting the change in absolute albedo units (e.g., +0.10) usually negates this problem and is what I usually advise doing.

Thank you for the clarification. We have used the absolute difference in the manuscript and clarified this with a sentence as follows (line 370 - 371).

"The percentage change in albedo was calculated in absolute terms as the change between albedo values with a pollutant (BC or dust or both) and a reference albedo value with zero pollutants (zero BC and dust concentration)."

**3.** With regard to the possibility of dust contamination of BC/OC measurements (an issue raised in my first review), the following addition to the paper is quoted several times: "According to our understanding the analysis method and amount of dust loading on the sample can also alter OC/BC ratios." I am glad to see the authors acknowledge this potential source of contamination, but I suggest they elaborate on this with a couple of additional sentences, for the benefit of readers. Specifically, I suggest explaining why dust can result in over-estimations of BC/OC burdens, citing one or papers to this effect, and if possible mentioning the potential magnitudes of overestimation that may occur.

We have added some sentences in the revised manuscript describing the impact of dust on OC/BC analysis as shown below (lines 303–314).

"The method used for analysis and the amount of dust loading on the sample can also affect the OC/BC ratio, as can the presence of metal oxides and calcium carbonate. High iron oxide concentrations can cause BC to pre-oxidize or drop off the filter, while calcium carbonate can be wrongly identified as BC. Laboratory studies have shown that the presence of metal oxides in aerosol samples can alter the OC/BC ratio either by enhancing OC charring or by lowering the BC oxidation temperature (Wang et al., 2010), while higher fractions of metal oxide can increase BC divergence across the thermal optical protocols (Wu et al., 2016). Dust can lead to a greater decrease in optical reflectance during the 250°C heating stage in the thermal/optical method, and thus an incorrect OC/BC ratio (Wang et al., 2012). Carbon detected by the flame ionization detector (FID) before the optical signal attains the initial value is defined as OC and that detected after is defined as BC; dust on the filter results in the FID division being postponed or inefficient, and thus OC being overestimated and BC underestimated or even negative (Wang et al., 2012). Wang et al. (2012) provides a more detailed discussion of OC/BC ratios derived using the thermal optical method."

**4.** The following sentence was added to the revised manuscript (quoted in the response to reviewers): "Warming role of OC was also not included in the current research, which was low but significant in several regions (Yasunari et al. 2015)." It is not clear to me what this means, so I request that this be clarified. Is it meant that albedo reduction from OC was not quantified here?

Yes, the purpose of the sentence was to show that the albedo reduction from organic carbon was not quantified in the present research. The sentence has been modified as shown below (line 624–627).

[revised manuscript text omitted]